# IDH1 mutation impairs antiviral response and potentiates oncolytic virotherapy in glioma

Xueqin Chen[1,2,12], Jun Liu ®[1,12], Yuqin Li[1,12], Yuequan Zeng[1], Fang Wang[1], Zexiong Cheng[1], Hao Duan ®[3], Guopeng Pan[1], Shangqi Yang[1], Yuling Chen[4,5], Qing Li[1], Xi Shen[1], Ying Li[1], Zixi Qin[1], Jiahong Chen[1,6], Youwei Huang[1], Xiangyu Wang[1], Yuli Lu[1,7], Minfeng Shu ®[4,5], Yubo Zhang[1], Guocai Wang[8], Kai Li[9], Xi Lin[1] ✉, Fan Xing[10] ✉ & Haipeng Zhang ®[1,11] ✉

IDH1 mutations frequently occur early in human glioma. While IDH1 mutation has been shown to promote gliomagenesis via DNA and histone methylation, little is known regarding its regulation in antiviral immunity. Here, we discover that IDH1 mutation inhibits virus-induced interferon (IFN) antiviral responses in glioma cells. Mechanistically, D2HG produced by mutant IDH1 enhances the binding of DNMT1 to *IRF3/7* promoters such that IRF3/7 are downregulated, leading to impaired type I IFN response in glioma cells, which enhances the susceptibility of gliomas to viral infection. Furthermore, we identify DNMT1 as a potential biomarker predicting which IDH1mut gliomas are most likely to respond to oncolytic virus. Finally, both D2HG and ectopic mutant IDH1 can potentiate the replication and oncolytic efficacy of VSVΔ51 in female mouse models. These findings reveal a pivotal role for IDH1 mutation in regulating antiviral response and demonstrate that IDH1 mutation confers sensitivity to oncolytic virotherapy.

Hotspot mutations of isocitrate dehydrogenase type 1 (*IDH1*) are an early and defining event in the development of a subgroup of gliomas. IDH1 mutations generate a gain-of-function conversion of α-ketoglutarate (αKG) to D-2-hydroxyglutarate (D2HG) that in turn inhibits DNA histone demethylases, resulting in hypermethylation-associated epigenetic tumor cell reprogramming[1]. Although its role in tumorigenesis is well established, it is unknown whether IDH1 mutation is involved in antiviral immunity.

[1]Department of Pharmacology, School of Medicine, Jinan University, 510632 Guangzhou, Guangdong, China. [2]Shanghai Institute of Hematology, State Key Laboratory of Medical Genomics, National Research Center for Translational Medicine (Shanghai), Ruijin Hospital affiliated with Shanghai Jiao Tong University School of Medicine, 200025 Shanghai, China. [3]Department of Neurosurgery/Neuro-oncology, State Key Laboratory of Oncology in South China, Sun Yat-sen University Cancer Center, 510060 Guangzhou, Guangdong, China. [4]Department of Pharmacology, School of Basic Medical Sciences, Shanghai Medical College, Fudan University, 200032 Shanghai, China. [5]Ministry of Education (MOE) & Ministry of Health (MOH) Key Laboratory of Medical Molecular Virology, School of Basic Medical Sciences, Shanghai Medical College, Fudan University, 200032 Shanghai, China. [6]The Institute of Cardiovascular Sciences, School of Basic Medical Sciences, Peking University, 100191 Beijing, China. [7]Shantou Centre for Disease Control and Prevention, 515000 Shantou, Guangdong, China. [8]Institute of Traditional Chinese Medicine & Natural Products, Guangdong Province Key Laboratory of Pharmacodynamic Constituents of TCM and New Drugs Research, College of Pharmacy, Jinan University, 510632 Guangzhou, Guangdong, China. [9]Guangdong Research Institute of Gastroenterology, The Sixth Affiliated Hospital of Sun Yat-sen University, 510655 Guangzhou, Guangdong, China. [10]Medical Research Institute, Guangdong Provincial People's Hospital (Guangdong Academy of Medical Sciences), Southern Medical University, 510080 Guangzhou, Guangdong, China. [11]Key Laboratory of Viral Pathogenesis & Infection Prevention and Control (Jinan University), Ministry of Education, 510632 Guangzhou, Guangdong, China. [12]These authors contributed equally: Xueqin Chen, Jun Liu, Yuqin Li. ✉e-mail: jnu_linxi@hotmail.com; xingfan@gdph.org.cn; zhanghp@jnu.edu.cn

Oncolytic viruses (OVs) are self-amplifying cancer biotherapeutics that can replicate in and kill tumor cells while sparing normal cells[2]. Oncolytic virotherapy is a potentially attractive strategy for the treatment of gliomas. Recently, a two-center phase I trial reported that patients with progressive pediatric high-grade gliomas who received intratumoral oncolytic herpes simplex virus-1 plus 5 Gy radiation seemed to have good clinical course and show immune activation in post-treatment tissues[3].

Clinical trials have showcased that oncolytic virus only works in a subset of patients, and an improved understanding of the determinants of their efficacy is needed[4]. It is widely accepted that tumor selectivity of oncolytic viruses by aberrant interferon signaling within tumor cells[5]. For example, glioblastoma (GBM) susceptibility to Newcastle Disease virus (NDV) is dependent on the loss of type I IFN genes at chromosome 9p21[6,7]. Cancer cells appear to sacrifice their antiviral defense mechanism to acquire many of the attributes of malignant phenotype[8]. Previous studies have demonstrated that genetic mutation that occur in cancer created the optimal cellular state for virus replication[5,9], which promotes us to investigate if IDH1 mutation in glioma confers sensitivity to oncolytic virus.

Retinoic acid-inducible gene I (RIG-I) like receptors (RLRs) are key sensors of virus infection[10,11]. Once activated, the adaptor proteins of these receptors activate the downstream protein kinases TBK1 and IKKs, which subsequently activate the transcription factor IRF3/7 and NF-κB, resulting in the production of type I interferons (IFN-I) and other antiviral or immunoregulatory factors[12]. IRF3 and IRF7 act as the key signal transcription factors responsible for the production of IFN-I and are essential for the establishment of innate immunity[13]. Until now, knowledge regarding the link between oncogenes and IRF3/7 remains limited.

Here, we discover a previously unknown function for IDH1 mutation in antiviral immunity. We show that D2HG, the product of mutant IDH1, downregulates the expression of IRF3/7 in a DNMT1-dependent manner. Consequently, either ectopically expressed mutant IDH1 or D2HG can enhance viral replication and therapeutic efficacy of VSVΔ51. Our study reveals an important additional function for IDH1 mutation in regulating IRF3/7 expression and could help explain the enhanced susceptibility of IDH1mut gliomas to oncolytic viruses.

## Results

### IDH1 mutation impairs IFN antiviral responses in glioma

To address whether IDH1 mutation is involved in antiviral immunity, we first analyzed the publicly available transcriptomic datasets from 3 independent clinical cohorts, including GEO and TCGA, and found that IFN signaling pathways were downregulated in IDH1mut gliomas versus IDH1wt gliomas (Fig. 1a, b). In addition, gene ontology analysis of the downregulated genes between IDH1mut samples and IDH1wt samples from GSE109857 revealed a highly significant enrichment of genes related to innate immune responses, such as granulocyte chemotaxis, antimicrobial humoral response, and regulation of B cell proliferation (Supplementary Fig. 1a).

To confirm that IDH1 mutation can regulate antiviral immune responses, we performed RNA sequencing (RNA-seq) in cells expressing IDH1(R132H) or a control vector upon VSVΔ51 infection. Our data identified 7 core signaling pathways enriched in three groups of comparisons (Fig. 1c). Notably, IFN signaling pathways were suppressed by exogenous expression of IDH1(R132H) (Fig. 1d and Supplementary Fig. 1b). As expected, VSVΔ51 infection activated IFN antiviral responses in glioma cells, while doxycycline-induced IDH1(R132H) appeared to inhibit VSVΔ51-induced activation of IFN signaling pathways (Fig. 1d, e and Supplementary Fig. 1c, d). Moreover, we found that the upregulated expression of IFN-stimulated genes (ISGs) induced by VSVΔ51 was abrogated by doxycycline-induced IDH1(R132H) to varying degrees (Fig. 1f). Together, these results

suggest that IDH1 mutation can impair antiviral responses, which indicates a critical role for IDH1 mutation in regulating antiviral immunity.

### IDH1 mutation renders glioma cells sensitive to VSVΔ51

Considering that IDH1 mutation suppresses IFN antiviral responses, we next investigated if IDH1 mutation could affect the replication and oncolysis of oncolytic viruses. Fluorescence microscopy showed a significantly enhanced replication of VSVΔ51 in cells expressing IDH1(R132H) compared to that in the control group (Fig. 2a). Consequently, the enhanced viral replication as indicated by green fluorescent protein (GFP) expression led to stronger inhibition of cell growth in cells expressing IDH1(R132H) than that in the control group (Fig. 2b, c). Furthermore, doxycycline-induced IDH1(R132H) enhanced the production of viral protein VSV-G (Fig. 2d). Single-step growth curves also demonstrated that IDH1(R132H) promoted viral replication in glioma cells (Fig. 2e). Similar results were observed in patient-derived GBM02 cell line and GL261 cell line (Fig. 2f–i and Supplementary Fig. 2a).

To further explore the effect of IDH1 mutation on viral replication in vivo, we performed tissue virus titration in a bilaterally tumor model (Fig. 2j). Consistent with our observation in vitro, immunofluorescence analyses and viral titers detection demonstrated that IDH1 mutation enhanced VSVΔ51 replication in vivo (Fig. 2k, l and Supplementary Fig. 2b).

### D2HG enhances VSVΔ51 replication and oncolysis

IDH1 normally converts isocitrate (IC) to α-ketoglutarate (αKG)[14]. IDH1 mutant induces a neomorphic enzymatic function that catalyzes the conversion of αKG to the structurally similar 2-hydroxyglutarate (2HG)[15] (Fig. 3a). We therefore tested if these metabolites have an impact on the viral replication of glioma cells. Fluorescence microscopy showed that neither IC nor αKG had effect on viral replication, whereas D2HG markedly enhanced VSVΔ51 replication in four glioma cell lines (Fig. 3b). Moreover, viral protein and titers were significantly increased in cells pretreated with D2HG (Fig. 3c, d). Furthermore, our data demonstrated that D2HG significantly enhanced the replication of Zika virus (ZIKV) and Herpes Simplex Virus type I (HSV-1) in glioma cells, thus indicating that the effect of D2HG also apply to other oncolytic viruses (Supplementary Fig. 3).

We next investigate if intracellular D2HG levels affect viral replication. LN-229 cells were treated with increasing concentrations of D2HG, followed by VSVΔ51 infection. Our data showed that D2HG promoted VSVΔ51 replication in a dose-dependent manner (Fig. 3e–g). We then detected the concentrations of D2HG, and found that intracellular D2HG levels correlated with viral replication (Fig. 3h, i). Furthermore, our data demonstrated that combined D2HG with VSVΔ51 treatment resulted in a stronger inhibition of cell viability (Fig. 3j). VSVΔ51, like other oncolytic viruses, has been reported to induce immunogenic cell death (ICD)[16]. We thus asked if the increased viral replication induced by D2HG could induce stronger ICD. We observed that VSVΔ51 induced ICD in glioma cells, which could be further enhanced when combined with D2HG, suggesting that D2HG can promote ICD induced by VSVΔ51 in GBMs (Fig. 3k). These results suggest that D2HG produced by mutant IDH1 might play key roles in promoting virus replication and oncolysis.

### D2HG promotes VSVΔ51 replication by suppressing IRF3/7 expression

RNA virus could be recognized by RIG-I-like receptors (RLRs), including retinoic acid-inducible gene I (RIG-I) and melanoma differentiation-associated gene 5 (MDA5) in the cytoplasm[17]. RIG-I and MDA5 could interact with the adapter mitochondrial antiviral signaling protein (MAVS), thereby activating TANK-binding kinase 1 (TBK1), which phosphorylates IRF3/7 and induces its activation and dimerization[10].

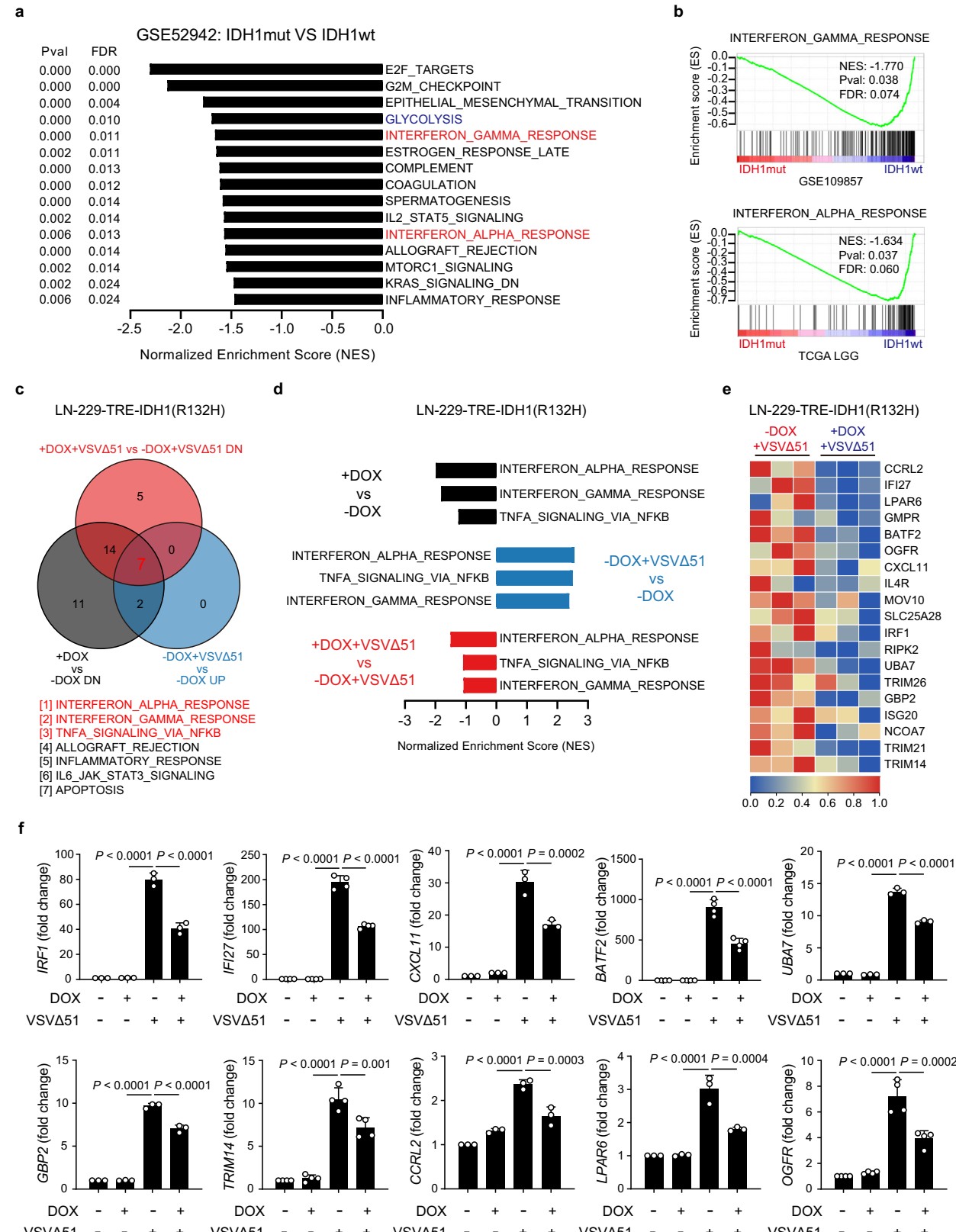

**Fig. 1 | IDH1 mutation inhibits IFN antiviral responses in glioma. a** Gene set enrichment analysis (GSEA) of downregulated pathways in IDH1mut gliomas versus IDH1wt gliomas from GEO. Pval, *P* value; FDR, false discovery rate. **b** GSEA showing interferon gamma response and interferon alpha response signatures in IDH1mut gliomas versus IDH1wt gliomas from GEO and TCGA. NES, normalized enrichment score; Pval, *P* value; FDR, false discovery rate. **c**–**e** RNA-seq analysis of cells infected with or without VSVΔ51 (MOI = 1) for 12 h in the presence or absence of doxycycline (DOX). **c** Venn diagram showing the shared signaling pathways identified by GSEA in three groups of comparisons. DN, downregulated; UP, upregulated. **d** The top 3 signaling pathways in the three groups of comparisons. **e** Heatmap showing the expression of signature genes related to interferon alpha response. **f** LN-229-TRE-IDH1(R132H) cells were infected with or without VSVΔ51 (MOI = 1) in the presence or absence of doxycycline for 12 h. qRT-PCR analysis assessing expression of IFN-stimulated genes. *n* = 3 or 4 biological replicates. Statistical significance was determined using one-way ANOVA in **f**. Data represent the mean ± SD. Source data are provided in the Source Data file.

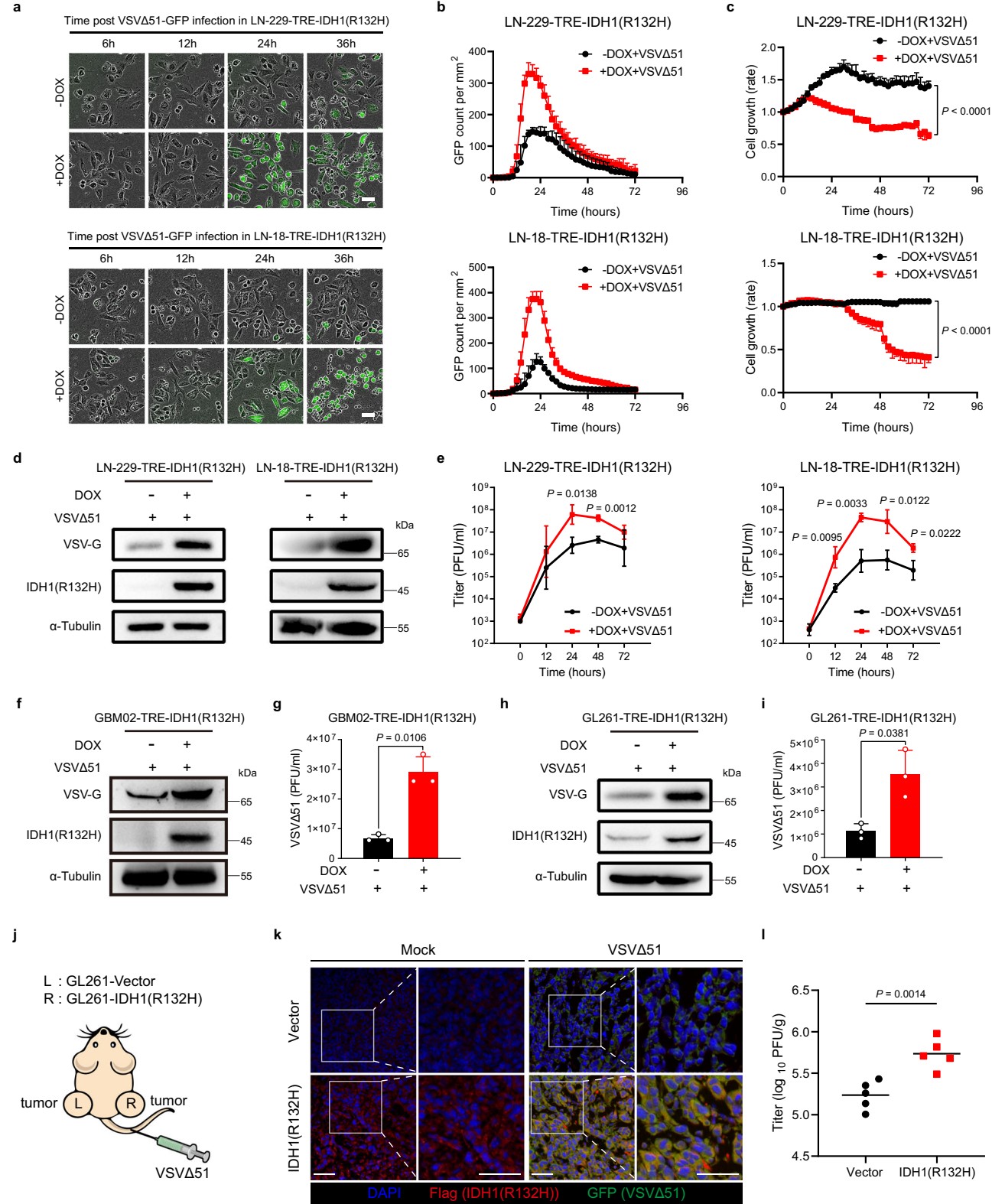

Finally, IRF3 and IRF7 bind to the promoter region of *IFNA* or *IFNB*, inducing type I interferon (IFN) responses and establishing an antiviral state[13,18] (Fig. 4a). The experiments described in this section were undertaken with the intent of testing the role of IDH1 mutation in cellular antiviral responses.

We first investigated whether IDH1 mutation or D2HG influences the status of RLR pathway. As shown in Fig. 4b, VSVΔ51 infection significantly activated RLR signaling pathway. Of note, doxycycline-induced IDH1(R132H) significantly inhibited the upregulated expression of ISGs (including RIG-I, MDA5, and STAT1) and phosphorylation of IRF3/7 induced by VSVΔ51. It is worth noting that IDH1(R132H) downregulated the total protein levels of IRF3/7 in cells either with or without VSVΔ51 infection (Fig. 4b and Supplementary Fig. 4a). Similar results were obtained using D2HG, which obviously suppressed the total protein levels of IRF3 and IRF7 (Fig. 4c and Supplementary Fig. 4b).

**Fig. 2 | IDH1 mutation enhances VSVΔ51 replication in glioma. a** Cells were pretreated with or without doxycycline (DOX) for 48 h, followed by VSVΔ51–GFP infection (MOI = 0.01 for LN-229, MOI = 0.1 for LN-18) for the indicated times. Phase-contrast and fluorescence microscopy images were captured. Representative images of $n = 3$. **b** Green fluorescent signals were measured, counted and analyzed in cells treated as in **a**. $n = 4$ biological replicates. **c** Relative cell growth rate was measured in cells treated as in **a**. $n = 4$ biological replicates. **d** Cells were infected with VSVΔ51 (MOI = 0.01 for LN-229, MOI = 0.1 for LN-18) for 24 h in the presence or absence of DOX, viral protein VSV-G was analyzed by western blot. **e** Cells were infected with VSVΔ51 (MOI = 0.01 for LN-229, MOI = 0.1 for LN-18) for the indicated times in the presence or absence of DOX, single-step growth analyses were conducted. $n = 3$ biological replicates. **f–i** Cells were infected with VSVΔ51 (MOI = 0.01 for GBM02, MOI = 1 for GL261) for 24 h in the presence or absence of DOX. **f, h** Viral protein VSV-G was analyzed by western blot. **g, i** Corresponding viral titers in supernatants were determined. $n = 3$ biological replicates. **j–l** The bilaterally implanted mice were treated with VSVΔ51 ($3 \times 10^7$ PFU) by intravenous injection, and tumors were resected at 24 h after injection. **j** Schematic illustration of the bilaterally subcutaneous transplantation model. **k** Immunofluorescence staining was used to evaluate the expression of GFP (reporter gene for VSVΔ51) and Flag-IDH1(R132H). Representative images of $n = 3$. **l** Subcutaneous tumor titers are shown for animals sacrificed 24 h after virus administration. $n = 5$ mice per group. Statistical significance was determined using two-way ANOVA in **c**, or two-tailed Student's $t$-test in **e, g, i, l**. Data represent the mean ± SD. Scale bar, 50 μm. Source data are provided in the Source Data file.

We next explored the importance of IRF3/7 in IDH1-mutation-mediated antiviral responses. Our data showed that ectopic expression of RIG-I, MDA5, MAVS, or TBK1 was not sufficient to inhibit the elevated replication of VSVΔ51 induced by D2HG in glioma cells (Supplementary Fig. 4c, d). In contrast, the increased viral protein and titer were significantly abrogated by exogenous IRF3/7, indicating an essential role of IRF3/7 in IDH1-mutation-regulated antiviral immunity (Fig. 4d). Furthermore, we found that IDH1(R132H) decreased the transcription of *IRF3/7* and downregulated the production of IFN-α and IFN-β in glioma cells (Fig. 4e, f and Supplementary Fig. 4e, f). Meanwhile, we observed that D2HG cannot further promote the replication in IRF3-knockdown or IRF7-knockdown glioma cells (Fig. 4g–l). Finally, we examined the protein expression of IRF3 in tumor tissues from 122 glioma patients, and found that the relative IRF3 expression was significantly lower in IDH1mut gliomas compared with IDH1wt gliomas (Fig. 4m). Collectively, our results reveal that D2HG produced by IDH1 mutation inhibits antiviral response by suppressing IRF3/7 expression.

## DNMT1 is required for D2HG-mediated downregulation of IRF3/7

D2HG has been thought to competitively inhibit αKG-dependent dioxygenases, such as ten-eleven translocation 2 (TET2) and histone lysine demethylases (KDMs), leading to the demethylation of DNA and histone respectively[19–21]. Given that the downregulated expression of IRF3/7 induced by IDH1 mutation appears to occur at transcriptional level, we asked whether D2HG could stimulate the hypermethylation of the *IRF3/7* promoters, which in turn leads to transcriptional silencing of IRF3/7. However, knockdown of KDM5C or TET2 did not mimic the capacity of D2HG to increase VSVΔ51 replication or inhibit the antiviral factors (Supplementary Fig. 5a–d). Meanwhile, silencing *KDM5C* or *TET2* cannot promote the oncolytic activity of VSVΔ51 (Supplementary Fig. 5e).

In mammalian cells, DNA methylation is carried out by three DNA methyltransferases (DNMTs): DNMT1; DNMT3a; and DNMT3b. DNMT1 is the most abundant DNMT in mammalian cells and is responsible for routine methylation maintenance[22,23]. A recent study reported that D2HG can bind directly to DNMT1 and regulate DNA methylation[24], which promotes us to investigate if DNMT1 participates in the suppression of IRF3/7 mediated by D2HG. First, knockdown of DNMT1 dramatically inhibited the replication of VSVΔ51 in glioma cells expressing IDH1(R132H) (Fig. 5a). Moreover, knockdown of DNMT1 promoted the transcription of *IRF3* in cells either with or without VSVΔ51 infection (Fig. 5b). Consistent with earlier results, D2HG was able to suppress IRF3/7 expression, leading to increased viral replication (Fig. 5c). Meanwhile, DNMT1-knockdown abrogated D2HG-induced IRF3/7 downregulation, and blocked VSVΔ51 replication (Fig. 5c, d). On the contrary, overexpression of DNMT1 promoted VSVΔ51 replication and suppressed IRF3/7 expression in glioma cells (Fig. 5e–g). We observed that exogenous expressed DNMT1 inhibited transcription from the *IRF3* promoter as measured by luciferase activity (Fig. 5h). Finally, Chromatin immunoprecipitation (ChIP) assay and ChIP-quantitative PCR (ChIP-qPCR) analyses demonstrated that

D2HG increased the association of DNMT1 with *IRF3/7* promoters, whereas, it did not affect the affinity of binding between DNMT1 and the *TBK1* or *IFNA1* promoter (Fig. 5i, j and Supplementary Fig. 5f, g). Taken together, these data indicate that D2HG's effects on DNMT1 and IRF3/7 that render IDH1mut glioma cells susceptible to virus infection (Fig. 5k).

Indeed, we found that D2HG was able to enhance VSVΔ51 oncolysis in a subset of glioma cell lines including LN-229, LNZ308, LN-18, and U118, as well as two patient-derived GBM cell lines (GBM01 and GBM02). However, it has no effect on the oncolytic activity of VSVΔ51 in two glioma cell lines (U251 and U87), one patient-derived GBM cell line GBM03 and primary human astrocytes (HA) (Supplementary Fig. 6a). We detected the protein levels of DNMT1 and calculated the relative difference in area under the curve (AUC) in the ten cell lines (Fig. 5l and Supplementary Fig. 6b), and found that expression of DNMT1 correlates with enhanced cell killing by the combination treatment with VSVΔ51 and D2HG (Fig. 5m). That is, high DNMT1 expression was associated with higher sensitivity to VSVΔ51/D2HG treatment. Thus, our results suggest that DNMT1 expression may be a useful biomarker to identify subsets of IDH1mut gliomas that are sensitive or resistant to oncolytic virotherapy.

## Both D2HG and ectopic IDH1 mutant enhance VSVΔ51 therapeutic efficacy in vivo

To determine whether D2HG could improve the therapeutic potential of VSVΔ51 in vivo, mice with subcutaneous GL261 tumors were treated with (i) intravenous VSVΔ51, (ii) intraperitoneal D2HG injection, or (iii) a combination of the two (Fig. 6a). As the data showed, the combined treatment markedly reduced tumor burden compared with monotherapies (Fig. 6b). At the endpoint, the levels of cleaved-caspase-3, Ki-67, VSV-G, and IRF3 were further examined in the subcutaneous xenograft tumor sections by immunohistochemistry (IHC) staining. We observed that combination treatment increased apoptosis and inhibited tumor cell proliferation (Fig. 6c). Meanwhile, D2HG abrogated the upregulation of IRF3 induced by VSVΔ51, thus resulting in increased viral replication (Fig. 6c). Furthermore, histological analyses of vital tissues displayed no significant abnormal pathological change in the combined treatment group (Fig. 6d).

To investigate whether mutant IDH1 could enhance the susceptibility of glioma cells to VSVΔ51, we orthotopically transplanted GL261-TRE-IDH1(R132H) cells into immunocompetent mice (Fig. 6e). Strikingly, mice that received doxycycline diet plus VSVΔ51 survived longer than those were untreated or received single treatments (Fig. 6f). Histopathology examination showed that doxycycline diet plus VSVΔ51 effectively restricted tumor growth compared with monotherapies alone (Fig. 6g). Similar to the IHC staining results mentioned above, VSVΔ51, in combination with doxycycline diet, (i) increased apoptosis, (ii) inhibited tumor cell proliferation, (iii) promoted viral replication, and (iv) enhanced infiltration of NK cells (Fig. 6h).

Considering oncolytic viruses can induce a potent, systemic, and potentially durable antitumor immunity[25,26], we hypothesized that the enhanced replication of VSVΔ51 induced by IDH1 mutation might

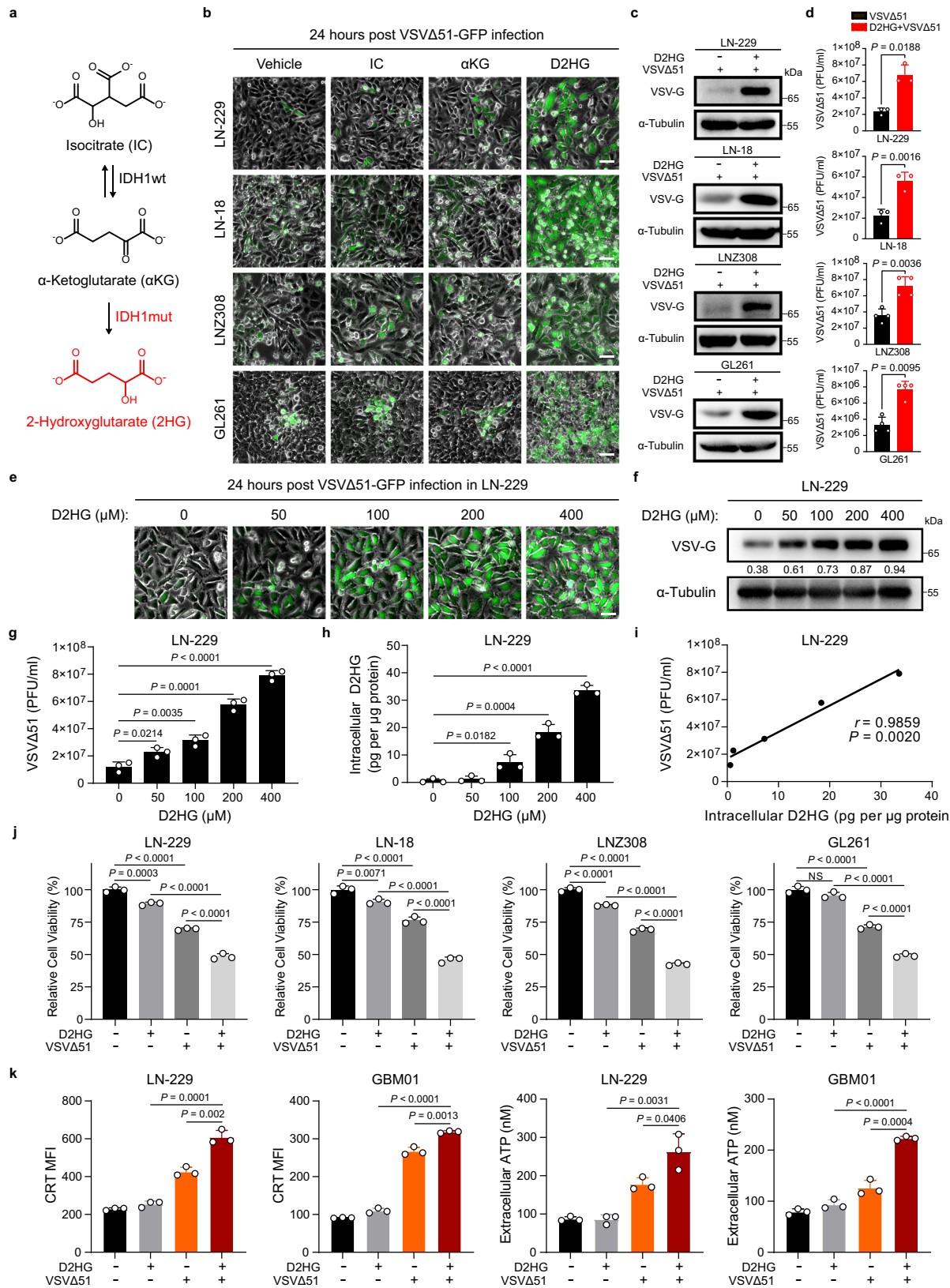

stimulate stronger antitumor immunity. First, we observed that doxycycline diet plus VSVΔ51 treatment led to enhanced infiltration of DC cells (Fig. 6i and Supplementary Fig. 7a, b), which can process and present tumor antigens to T cells for initiating anti-tumor response. Consequently, combination treatment increased the number of CD8+ T cells, with elevated activation marker (granzyme B), as well as

reduced regulatory T cells (Tregs) in tumor tissues (Fig. 6i and Supplementary Fig. 8). Moreover, our data showed that doxycycline diet plus VSVΔ51 treatment led to increased percentage of VSV $N_{52-59}$ tetramer positive CD8+ T cells compared with VSVΔ51 treatment (Fig. 6i and Supplementary Fig. 7c), indicating that the increased percentage of CD8+ T cells were virus specific.

**Fig. 3 | D2HG augments viral replication and oncolytic activity of VSVΔ51 in glioma. a** Metabolism of 2HG. **b** Cells were pretreated with vehicle, IC (4 mM), αKG (4 mM), or D2HG (400 μM) for 48 h, followed by VSVΔ51–GFP infection (MOI = 0.01 for LN-229, MOI = 0.1 for LN-18 and LNZ308, MOI = 1 for GL261) for 24 h, after which the phase-contrast and fluorescence microscopy images were captured. Representative images of *n* = 3. **c** Cells were pretreated with D2HG (400 μM) for 48 h and subsequently infected with VSVΔ51 (MOI = 0.01 for LN-229, MOI = 0.1 for LN-18 and LNZ308, MOI = 1 for GL261) for 24 h, VSV-G was analyzed by western blot. **d** Viral titers were determined in cells treated as in **c**. *n* = 3 or 4 biological replicates. **e**–**i** Cells were pretreated with vehicle or increasing concentrations of D2HG for 48 h, followed by VSVΔ51–GFP (MOI = 0.01) infection for 24 h. **e** Phase-contrast and fluorescence microscopy images were captured. Representative images of *n* = 3.

**f** Viral protein VSV-G was analyzed by western blot. **g** Corresponding viral titers in supernatants were determined. **h** Intracellular D2HG measurements in cells pretreated with vehicle or D2HG for 48 h. **i** Pearson's correlation coefficient (*r*) and the two-tailed *P* value of the linear correlation analysis performed between viral titers and intracellular D2HG levels. *n* = 3 biological replicates. **j** Relative cell viability in the cells treated with vehicle, D2HG (400 μM), VSVΔ51 (titers as in **c**), or a combination for 72 h. *n* = 3 biological replicates. NS not significant. **k** Cells were treated with vehicle, D2HG (400 μM), VSVΔ51 (MOI = 0.01 for LN-229, MOI = 1 for GBM01) or a combination for 24 h. CRT exposure and ATP secretion were detected and quantified. *n* = 3 biological replicates. Statistical significance was determined using two-tailed Student's *t*-test in **d**, **g**, **h**, **k**, or one-way ANOVA in **j**. Data represent the mean ± SD. Scale bar, 50 μm. Source data are provided in the Source Data file.

To further address if the combination treatment could activate the systemic tumor-specific cellular immunity, splenic lymphocytes were isolated from tumor-bearing mice receiving different treatments and co-cultured with GL261 cells. We found that the splenic lymphocytes from dual treatment group demonstrated higher cytotoxicity and secreted significantly more IFN-γ and TNF-α than those from VSVΔ51 treatment group (Fig. 6j), suggesting that there is a specific antitumor response.

We next analyzed if long-lasting protective anti-tumor immunity is formed upon doxycycline diet plus VSVΔ51 treatment. We observed that the combination treatment led to enhanced recall responses of central memory CD8+ T cells (T_CM) (Fig. 6i and Supplementary Fig. 7d). Meanwhile, mice that received combination treatment rejected rechallenged tumor (Fig. 6k). Collectively, these results demonstrate the generation of an immune memory after combination treatment.

## IDH1 mutation renders patient-derived GBMs sensitive to VSVΔ51

To investigate if IDH1mut GBMs are more suspectable to OV infection, we chose two IDH1wt patient-derived GBM cell lines (GBM02, GBM03) and two IDH1mut cell lines (GBM04, GBM05) to analyze the viral replication and antiviral responses (Fig. 7a). Our data showed that the capacity of viral replication in IDH1mut GBM cells was significantly stronger than that in IDH1wt GBM cells (Fig. 7b). Meanwhile, the expression of IRF3/7 was lower in IDH1mut GBM04 cells than that in IDH1wt GBM03 cells (Fig. 7c), indicating the antiviral responses are suppressed in IDH1mut GBMs.

We next determine whether D2HG and IDH1mutation could improve the therapeutic potential of VSVΔ51 in patient-derived GBMs. As our data showed, D2HG can enhance viral replication in GBM02 cells (Fig. 7d, e). Furthermore, we orthotopically transplanted GBM02-TRE-IDH1(R132H) cells into mice (Fig. 7f). Consistent with our observation in GL261 xenograft model, doxycycline diet plus VSVΔ51 led to longer survival and effectively restricted tumor growth compared with monotherapies alone (Fig. 7g, h). Similar to the IHC staining results mentioned above, VSVΔ51, in combination with doxycycline diet, (i) increased apoptosis, (ii) inhibited tumor cell proliferation, (iii) promoted viral replication, and (iv) enhanced infiltration of NK cells (Fig. 7i).

## Discussion

It has become abundantly clear that IFN signaling pathway components, such as IFN-α、IRFs、RIG-I, and STAT1, are often disrupted through multiple genetic mechanisms during tumorigenesis, most notably, mutation and DNA methylation[27–30]. For example, PTEN has been reported to positively regulate the import of IRF3 into the nucleus and innate immune responses, and PTEN-mutant mice were more susceptible to VSV infection than were their PTEN-wild-type counterparts[31]. Such oncogenic alterations drastically compromise the ability of tumor cells to combat viral infection by impairing viral sensing and promoting immune evasion. On this basis, oncolytic viruses can selectively infect and kill cancer cells by exploiting their impaired antiviral response[32,33].

Recent studies have reported that IDH1 mutation plays a critical role in shaping the immune microenvironment of glioma[34–38], while its role in the regulation of antiviral immune responses remains unclear. Notably, our data demonstrated that mutant IDH1 abrogated VSVΔ51-induced cellular antiviral responses, thus indicating a critical role for IDH1 mutation in antiviral immunity. We observed that either ectopic IDH1 mutant or its metabolite D2HG renders glioma cells susceptible to VSVΔ51 infection. Thus, our data indicate that patients with IDH1mut gliomas might be more sensitive to OV infection.

RLRs are RNA sensors localized in cytosol[11]. Through the analysis of RLR signaling pathway, we identified IRF3 and IRF7 as the key effectors of mutant IDH1 that are implicated in antiviral immune responses. IRF3/7 are master transcriptional factors that regulate type I interferon gene induction and innate immune defenses after virus infection[13]. We discovered that DNMT1 is critical for D2HG-induced IRF3/7 downregulation and may serve as a potential biomarker predicting which IDH1mut gliomas are most likely to respond to oncolytic viruses.

Our study does present some limitations. Here, we found that intraperitoneal injection of D2HG enhanced viral replication and improved therapeutic efficacy of VSVΔ51. However, the levels of xenograft D2HG should be detected to show the in vivo delivery rate of D2HG in the future. Although glioma xenografts carrying IDH1 mutations are very scarce, further investigation should be designed to compare the sensitivities of OVs in IDH1wt and IDH1mut xenografts.

Overall, our study reveals a hitherto unrecognized link between IDH1 mutation and IRF3/7 in antiviral immunity and provides a theoretical basis for clinical treatment of IDH1mut gliomas by oncolytic viruses.

## Methods

### Ethics statement

All animal experiments described in this study were approved by the Laboratory Animal Ethics Committee of Jinan University (IACUC no. 20210816-04; 20230205-07; 20230205-12). Tumor samples were collected with the patients' written informed consent and approved by the Institutional Ethics Committee for Clinical Research and Animal Trials of the Sun Yat-sen University Cancer Center (no. B2021-259-01).

### Analysis of expression and clinical data from public databases

Expression and clinical data of the glioma patient datasets (GSE52942 and GSE109857) was downloaded from NCBI Gene Expression Omnibus (GEO; www.ncbi.nlm.nih.gov/geo). Differential expression between IDH1wt gliomas and IDH1mut gliomas from patients was calculated using the limma R-package[39]. Expression and clinical data of The Cancer Genome Atlas (TCGA) lower-grade glioma (LGG) cohort was obtained from UCSC Xena (https://xenabrowser.net/datapages/). Gene Set Enrichment Analysis (GSEA) was used to analyze the signal pathway enrichment in different groups of samples. ClueGO, a Cytoscape App was used to select representative Gene Ontology terms

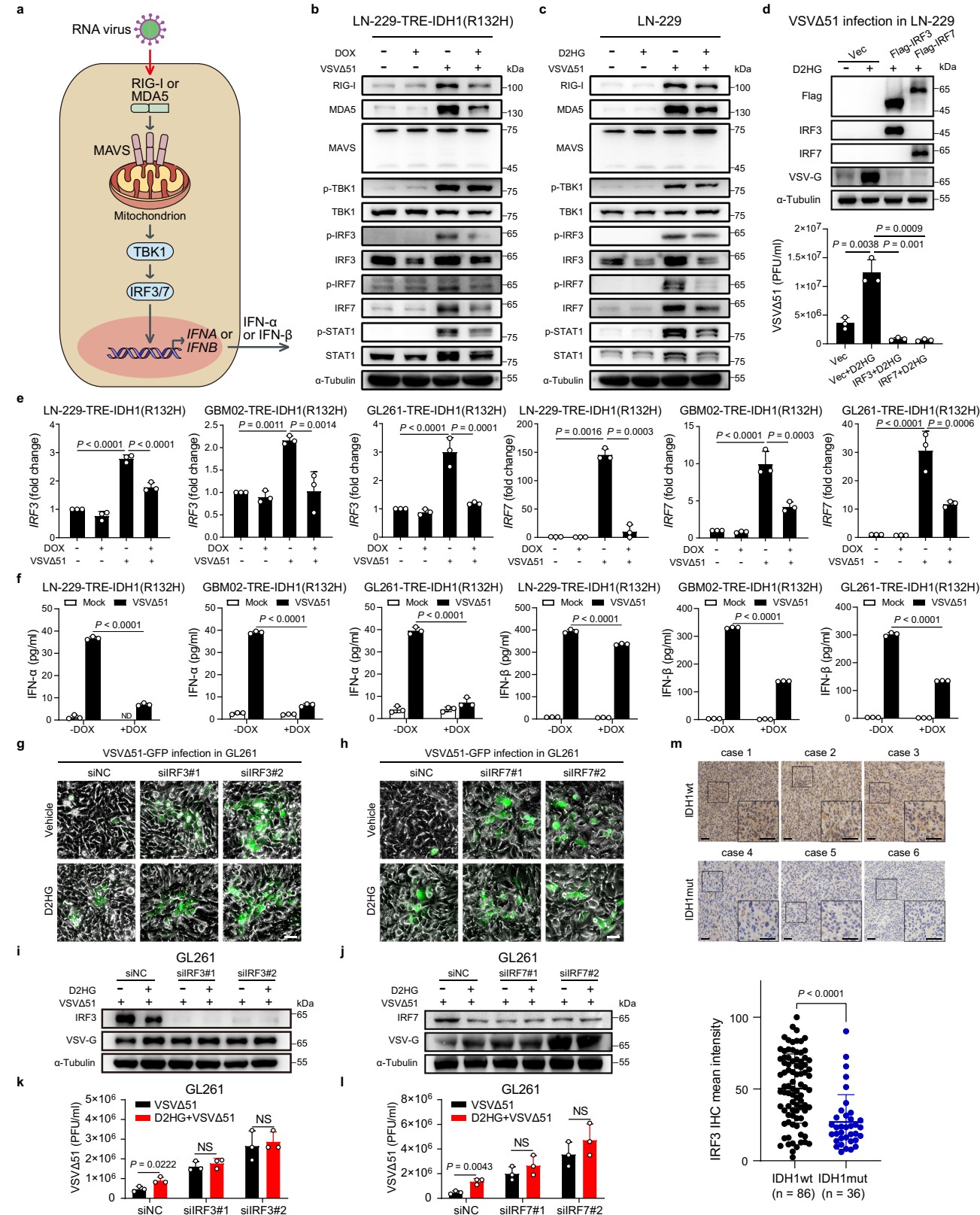

and pathways for one or multiple lists of genes/proteins and visualize them into functionally organized networks.

## Cell lines and viruses

GL261, U251, and U87 cells were generously gifted by Prof. Guangmei Yan (Zhongshan School of Medicine, Sun Yat-sen University). LN-18, LN-229, U118, LNZ308 cells were generously gifted by Prof. Qi Qi

(School of Medicine, Jinan University). Primary patient-derived GBM cells were generously gifted by Dr. Hao Duan (Sun Yat-sen University Cancer Center). GBM04 and GBM05 cell lines are IDH1mut, and other glioma cell lines and primary patient-derived GBM cell lines used in this study are IDH1wt. Primary patient-derived GBM cells were maintained in Dulbecco's modified Eagle medium/Nutrient Mixture F-12 (DMEM/F-12) supplemented with 10% fetal bovine serum (FBS) and 1% penicillin/

**Fig. 4 | D2HG enhances VSVΔ51 replication through inhibition of IRF3/7 expression. a** Schematic illustration of RLR signaling pathway. **b, c** Cells pretreated with doxycycline (DOX) or D2HG (400 μM) for 48 h and subsequently infected with VSVΔ51 (MOI = 1) for 12 h. Proteins were examined by western blot. **d** Cells were pretreated with D2HG (400 μM) for 48 h and then transfected with the indicated plasmids for 24 h, followed by infection with VSVΔ51 (MOI = 1) for 24 h. Proteins were examined by western blot, and corresponding viral titers in supernatants were determined. $n = 3$ biological replicates. **e** Cells were pretreated with DOX for 48 h and subsequently infected with VSVΔ51 (MOI = 1) for 12 h. qRT-PCR assessing expression of *IRF3* and *IRF7* mRNA. $n = 3$ biological replicates. **f** Cells pretreated with DOX for 48 h and subsequently infected with VSVΔ51 (MOI = 1) for 12 h, supernatants were collected and assayed by ELISA for IFN-α and IFN-β production.

$n = 3$ biological replicates. ND not detectable. **g–j** Cells were transfected with siRNA targeting IRF3 or IRF7 for 48 h and then treated with D2HG (400 μM) for 48 h, followed by VSVΔ51 infection (MOI = 10) for 24 h. **g, h** Phase-contrast and fluorescence microscopy images were captured. Representative images of $n = 3$. **i, j** Proteins were examined by western blot. **k, l** Corresponding viral titers in supernatants were determined. $n = 3$ biological replicates. NS not significant. **m** Representative images and statistical analysis of IRF3 immunohistochemical staining in human IDH1wt and IDH1mut gliomas. Representative images of $n = 3$. Statistical significance was determined using one-way ANOVA in **d, e**, two-way ANOVA in **f, k, l**, or two-tailed Student's *t*-test in **m**. Data represent the mean ± SD. Scale bar, 50 μm. Source data are provided in the Source Data file.

streptomycin. BHK-21 cells were maintained in Minimum Essential Medium (MEM) supplemented with 10% FBS and 1% penicillin/streptomycin. All other cell lines were maintained in Dulbecco's modified Eagle medium (DMEM) supplemented with 10% FBS and 1% penicillin/streptomycin. Primary human astrocytes (HA) were purchased from ScienCell Research Laboratories and cultured according to the instructions provided. The cell lines have been authenticated by the short tandem repeat (STR) assay and were confirmed to be without mycoplasma contamination. VSVΔ51 and ZIKV were provided by Prof. Fan Xing (Guangdong Provincial People's Hospital). HSV-1 was provided by Prof. Minfeng Shu (School of Basic Medical Sciences, Fudan University). VSVΔ51-GFP is a recombinant derivative of VSVΔ51 expressing jellyfish green fluorescent protein. VSVΔ51 was propagated in BHK-21 cells, ZIKV and HSV-1 were propagated in Vero cells. Virus titer was determined by TCID50 assay using BHK-21 cells and converted to PFU.

## RNA-seq and relative data analysis
To profile the gene expression differences, total RNA was extracted using TRIzol following the manufacturer's instructions and then sent to the HaploX Genomics Center for RNA-sequencing analysis performed by Illumina NovaSeq 6000 instrument. Gene Set Enrichment Analysis (GSEA) was used to analyze the signal pathway enrichment in different groups of samples. The Venn diagram and the heat map of normalized expression values were generated using TBtools.

## qRT-PCR
Total RNA was extracted using TRIzol (Life Technologies), and reverse transcription was performed from 3 μg total RNA using oligo(dt) and RevertAid Reverse Transcriptase (Thermo Scientific) according to the manufacturer's recommendation. qRT-PCR was performed with SuperReal PreMix SYBR Green (TIANGEN) using an CFX96 Real-Time PCR Detection System (Bio-Rad). All genes were normalized to *ACTB*. The primer sequences used are listed in Supplementary Table 1.

## Lentivirus production and infection
Negative control empty vector, pLenti-TRE-human IDH1(R132H)-EF1-rtTA3-Puro and pLenti-TRE-murine IDH1(R132H)-CBH-Tet-On@3G-BSR were purchased from OBiO Technology (Shanghai, China). Lentiviruses were produced in HEK-293T cells using the Lenti-Pac HIV Expression Packaging Kit according to the manufacturer's instructions. LN-229, LN-18, GL261 and GBM02 cells were introduced with lentivirus and 5 μg/ml polybrene for 4 h. Forty-eight hours after introduction, the cells were selected with 2 μg/ml puromycin or 2 μg/ml blasticidin for 7-10 days to establish stably expressing cell lines. After selection, 1 μg/ml doxycycline (DOX) was added to induce the expression of IDH1(R132H).

## IncuCyte live-cell imaging assay
Cellular morphology and magnitude of confluence were assessed in real-time using the IncuCyte Live-Cell Imaging Analysis System (Essen BioScience). LN-229-TRE-IDH1(R132H) cells and LN-18-TRE-

IDH1(R132H) cells were seeded in six-well plates at $1 \times 10^5$ cells per well for 48 h in the presence or absence of doxycycline (DOX). Thereafter, cells were infected with VSVΔ51-GFP and placed in the IncuCyte System. Confluency was measured by averaging the percentage of area that the cells occupied from more than 3 images of a given well every 2 h for 72 h. Green fluorescent signals were measured, and green-fluorescent cells were counted as virus-infected cells. The assay was performed according to the manufacturer's protocol.

## Antibodies and reagents
Antibodies used in this study are listed as follows: IDH1(R132H) (26081, NewEast, 1:1000), VSV-G (8G5F11, Kerafast, 1:1000 for IB, 1:100 for IHC), Flag (F1804, Sigma, 1:1000 for IB, 1:500 for IF), GFP (GB13227, Servicebio, 1:1500), RIG-I (3743, Cell Signaling Technology, 1:1000), MDA5 (5321, Cell Signaling Technology, 1:1000), MAVS (3993, Cell Signaling Technology, 1:1000), TBK1 (3504, Cell Signaling Technology, 1:1000), phosphorylated TBK1 (5483, Cell Signaling Technology, 1:1000), IRF3 (ab68481, Abcam, 1:1000 for IB, 1:500 for IHC), phosphorylated IRF3 (29047, Cell Signaling Technology, 1:1000), IRF7 (4920, Cell Signaling Technology, 1:1000), phosphorylated IRF7 (5184, Cell Signaling Technology, 1:1000), STAT1 (14994, Cell Signaling Technology, 1:1000), phosphorylated STAT1 (9167, Cell Signaling Technology, 1:1000), DNMT1 (NB100-56519, Novus, 1:1000 for IB, 1:150 for ChIP), Ki-67 (GB111499, Servicebio, 1:200), cleaved-caspase-3 (GB11532, Servicebio, 1:200), NKp44 (GB11615, Servicebio, 1:200), CD45 (103108, 103114, BioLegend, 1:200), CD3 (100330, BioLegend, 1:200), CD4 (100536, BioLegend, 1:200), CD8 (100752, BioLegend, 1:100), Foxp3 (320014, BioLegend, 1:100), Granzyme B (372214, BioLegend, 1:200), CD86 (105005, BioLegend, 1:200), CD11 (117327, BioLegend, 1:100), CD44 (103049, BioLegend, 1:200), CD62L (104436, BioLegend, 1:100), α-Tubulin (ARG65693, Arigo, 1:5000), and GAPDH (ARG65680, Arigo, 1:5000).

H-2K$^b$ VSV NP$_{52-59}$ RGYVYQGL (PE–labeled tetramer) was purchased from MBL. Zombie Red™ Fixable Viability Kit and True-Nuclear™ Transcription Factor Staining Kit were purchased from Biolegend. Doxycycline was purchased from Selleck. DL-isocitric acid trisodium salt hydrate was purchased from Macklin. Dimethyl α-ketoglutarate was purchased from Sigma. (2R)-2-hydroxyglutaric acid octyl ester sodium salt was purchased from Toronto Research Chemicals.

## D2HG concentration evaluation
D2HG Assay Kit (ab211070, Abcam) was utilized to evaluate the concentration of D2HG in cell lysates. The levels of D2HG were measured according to the manufacturer's instruction.

## Western blot analysis
Cells were lysed using the M-PER Mammalian Protein Extraction Reagent (Thermo Scientific), and sodium dodecyl sulfate–polyacrylamide gel electrophoresis was performed. Membranes were visualized on a chemiluminescence imaging system (Tanon, 4600, Shanghai, China) using Immobilon Western

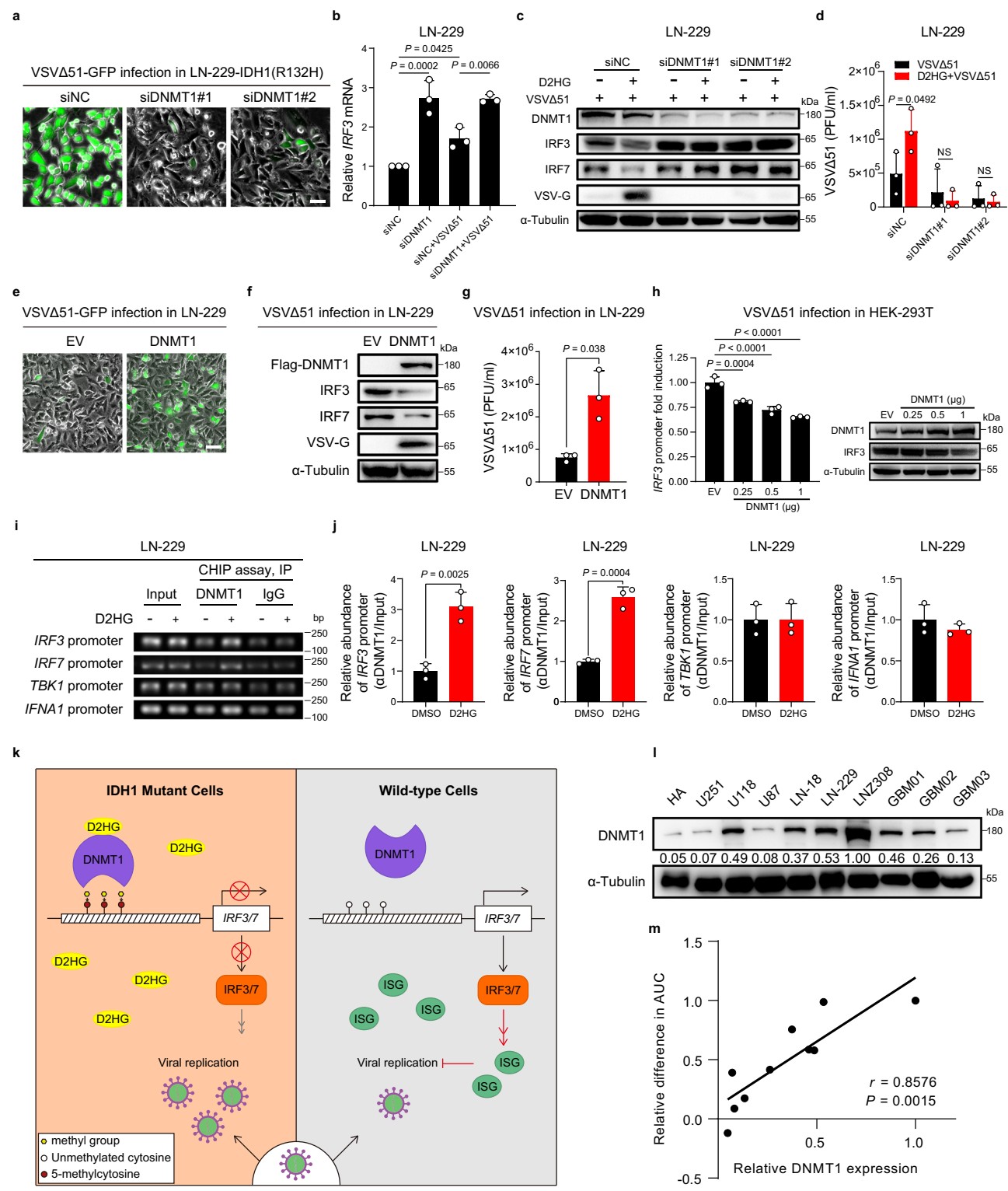

Chemiluminescent HRP Substrate (Millipore). A certain protein was quantified as the ratio of itself and α-Tubulin intensity bands by Image J software. Relative DNMT1 expression was calculated according to the formula: Ratio$_x$/Ratio$_{LNZ308}$.

### Detection of surface CRT and ATP release

Cells were stained with anti-CRT (12238, Cell Signaling Technology, 1:800), and then stained with the secondary antibody Alexa Fluor 488 goat anti-rabbit (4412, Cell Signaling Technology, 1:500). Surface CRT was detected by flow cytometry. Extracellular ATP was quantified using an ENLITEN ATP Assay System Bioluminescence Detection Kit (FF2000, Promega).

### Animal models

Four- to six-week-old female BALB/c-nu/nu mice and four-week-old female C57BL/6 mice were purchased from GemPharmatech Co. (Jiangsu, China). Animals were housed in SPF facilities in the experimental animal center of Jinan University. Mice were housed at an ambient temperature of 22–24 °C, humidity-controlled environment at 40–70% under a 12-h light/dark cycle with ad libitum access to water

**Fig. 5 | D2HG inhibits IRF3/7 expression by enhancing the association of DNMT1 with the *IRF3/7* promoters. a** Cells were pretreated with doxycycline (DOX) for 48 h and then transfected with siRNA targeting DNMT1 for 48 h, followed by VSVΔ51 infection (MOI = 1) for 24 h. Phase-contrast and fluorescence microscopy images were captured. Representative images of *n* = 3. **b** Cells were treated with siRNA targeting DNMT1 for 48 h, followed by VSVΔ51 infection (MOI = 1) for 12 h, and then processed for qRT-PCR to measure *IRF3* transcripts. *n* = 3 biological replicates. **c, d** Cells were transfected with siRNA targeting DNMT1 for 48 h and then treated with D2HG (400 μM) for 48 h, followed by VSVΔ51 infection (MOI = 1) for 24 h. **c** Western blot analysis of the indicated proteins. **d** Viral titers in supernatants. *n* = 3 biological replicates. NS, not significant. **e–g** Cells were transfected with empty vector (EV) or plasmid encoding DNMT1 for 24 h, followed by VSVΔ51 infection (MOI = 1) for 24 h. **e** Phase-contrast and fluorescence microscopy images.

**f** Western blot analysis of the indicated proteins. **g** Viral titers in supernatants. *n* = 3 biological replicates. **h** Dual-luciferase reporter assay (left) and immunoblot analysis of DNMT1 and IRF3 (right). *n* = 3 biological replicates. **i** Chromatin immunoprecipitation (ChIP) analysis in LN-229 cells treated with or without D2HG. **j** qRT-PCR analysis of the relative abundance of the indicated promoter segment in the ChIP assays as in **i**. *n* = 3 biological replicates. **k** Graphical model showing the role of IDH1 mutation-mediated IRF3/7 downregulation and OV sensitivity. **l** DNMT1 protein expression in different cell lines. Representative images of *n* = 3. **m** Pearson's correlation coefficient (*r*) and the two-tailed *P* value of the linear correlation analysis performed between relative difference in AUC and DNMT1 expression. Statistical significance was determined using one-way ANOVA in **b, h**, two-way ANOVA in **d**, or two-tailed Student's *t*-test in **g, j**. Data represent the mean ± SD. Scale bar, 50 μm. Source data are provided in the Source Data file.

and food. Animal care and handling procedures were in accordance with the Institutional Animal Care and Use Committee (IACUC) protocol and were approved by the Jinan University Institutional Review Board. According to the IACUC protocol, the maximal tumor burden is 1500 mm³ permitted by ethics committee. When the tumor volume was up to 1500 mm³, it was considered as the humane endpoint.

For the bilaterally subcutaneous transplantation model, GL261-vector or GL261-TRE-IDH1(R132H) cells (3 × 10⁶ cells/mouse) were inoculated subcutaneously into the hind flank of 4-week-old female BALB/c-nu/nu mice. The day after tumor implantation, doxycycline (DOX, 30 mg/kg/day) was given by intragastric (i.g.) injection every other day. After 14 days, palpable tumors developed (~50 mm³), VSVΔ51 (3 × 10⁷ PFU) was intravenously injected, and tumors were resected at 24 h after injection.

For the subcutaneous transplantation model, dissociated GL261 cells (1 × 10⁶) in 100 μl of PBS were inoculated subcutaneously into the hind flanks of 6-week-old female BALB/c-nu/nu mice. When palpable tumors have developed (~50 mm³) in ~2 weeks, the mice were randomly divided into 4 groups and treated with different agents. D-2-hydroxyglutarate (D2HG, 6 mg/kg/day) was administered via intraperitoneal (i.p.) injection on days 14–17. VSVΔ51 (1 × 10⁷ PFU/day) was administered by tail vein injection on days 15–17. Tumor length and width were measured every 2 days, and the volume was calculated according to the formula (length × width²)/2.

For the orthotopic glioblastoma model, dissociated glioma cells (3 × 10⁵ cells) in 5 μl of PBS were implanted stereotactically into the striatum (2.2 mm lateral from the bregma and 2.5 mm deep) of mice. 4-week-old female C57BL/6 mice were orthotopically transplanted with GL261-TRE-IDH1(R132H) cells. 6-week-old female BALB/c-nu/nu mice were orthotopically transplanted with GBM02-TRE-IDH1(R132H) cells. 4 days after tumor implantation, the mice were randomly divided into 4 groups and treated with different agents. For C57BL/6 mice, doxycycline (DOX, 30 mg/kg/day) was administrated via intragastric (i.g.) injection every other day on days 4–12. VSVΔ51 (1 × 10⁷ PFU/day) was injected via tail vein (i.v.) on days 10–12. For BALB/c-nu/nu mice, DOX-containing (3 mg/ml) drinking water was provided to the mice until day 21. VSVΔ51 (1 × 10⁷ PFU/day) was injected via tail vein (i.v.) on days 21–23.

For tumor rechallenge study, long-term surviving mice received combination treatment of DOX and VSVΔ51 were rechallenged with an increased dose of GL261-TRE-IDH1(R132H) cells (6 × 10⁵ cells) in the contralateral hemisphere. The age-matched naive mice were implanted with the same number of cells as the control group.

## Histological analysis
For immunofluorescence (IF), tumors were harvested and postfixed overnight in periodate-lysine-paraformaldehyde fixative at 4 °C. Sections were cut and rehydrated in PBS, permeabilized in 0.1% Triton X-100, blocked in 5% goat serum and incubated with primary antibodies including GFP (reporter gene of VSVΔ51) and Flag overnight at 4 °C. After the slides were washed in PBS, they were incubated with

secondary antibodies, counterstained with DAPI (Invitrogen), washed in 0.1% Triton X-100 and mounted using Prolong Diamond antifade (Invitrogen). The above assays were made in a blinded manner for pathologists. For immunohistochemistry (IHC), tumor or brain sections (4 μm) were dewaxed in xylene, hydrated in decreasing concentrations of ethanol, immersed in 0.3% H₂O₂-methanol for 30 min, washed with phosphate-buffered saline, and probed with monoclonal antibodies or isotype control at 4 °C overnight. After being washed, the sections were incubated with biotinylated goat anti-rabbit or anti-mouse IgG at room temperature for 2 h. Immunostaining was visualized with streptavidin/peroxidase complex and diaminobenzidine, and sections were then counterstained with hematoxylin. For hematoxylin and eosin (H&E) staining, vital tissues (including brain, heart, kidney, liver, lung, skeletal muscle, and spleen) or brain sections were dewaxed in xylene, hydrated in decreasing concentrations of ethanol and stained with H&E. The IHC and H&E sections were digitalized using a Zeiss Axio Scan.Z1 slide scanner and analyzed by Image J software. The data of IHC sections were analyzed on a cell-by-cell basis.

## Tissue virus titration
Tumor tissues were homogenized using gentleMACS Dissociator (Miltenyi Biotec). Briefly, tumors were weighed and collected in 500 μl of ice-cold 1×PBS and then transferred to gentleMACS M Tube. Tubes were attached upside down on the sleeve of the dissociator, and gentleMACS Program RNA_1 was initiated to collect the homogenate. Homogenate was centrifuged at 600×g for 10 min, and supernatant was collected and titered on BHK-21 cells to quantify infectious virus.

## Cell viability assay
Cells were seeded in 96-well plates at 3,000 cells per well in 0.1 ml medium. After treatment, 3-(4,5-dimethylthiazol-2-yl)−2,5-diphenyltetrazolium bromide (MTT) was added to the cells (1 mg/ml final concentration), and the cells were allowed to grow at 37 °C for another 3 h. MTT-containing medium was removed, and the MTT precipitate was dissolved in 100 μl DMSO. The optical absorbance was determined at 490 nm using a microplate reader (iMark, Bio-Rad).

## Plasmid construction and transfection
Prof. Fan Xing provided mammalian expression plasmids for RIG-I, MDA5, MAVS, TBK1, IRF3 and IRF7. The *IRF3* promoter dual-luciferase reporter plasmid and the mammalian expression plasmid for DNMT1 were purchased from Tsingke Biotechnology Co., Ltd. All constructs were verified by DNA sequencing. Lipofectamine 3000 (Invitrogen) was used for transfection of plasmids into cells.

## Detection of cytokine production
The production and secretion of IFN-α or IFN-β in cell supernatants was measured with a human IFN-α ELISA kit (BSEH-084, Biosharp), a human IFN-β ELISA kit (BSEH-274, Biosharp), a mouse IFN-α ELISA kit (BSEM-117, Biosharp) and a mouse IFN-β ELISA kit (BSEM-116, Biosharp).

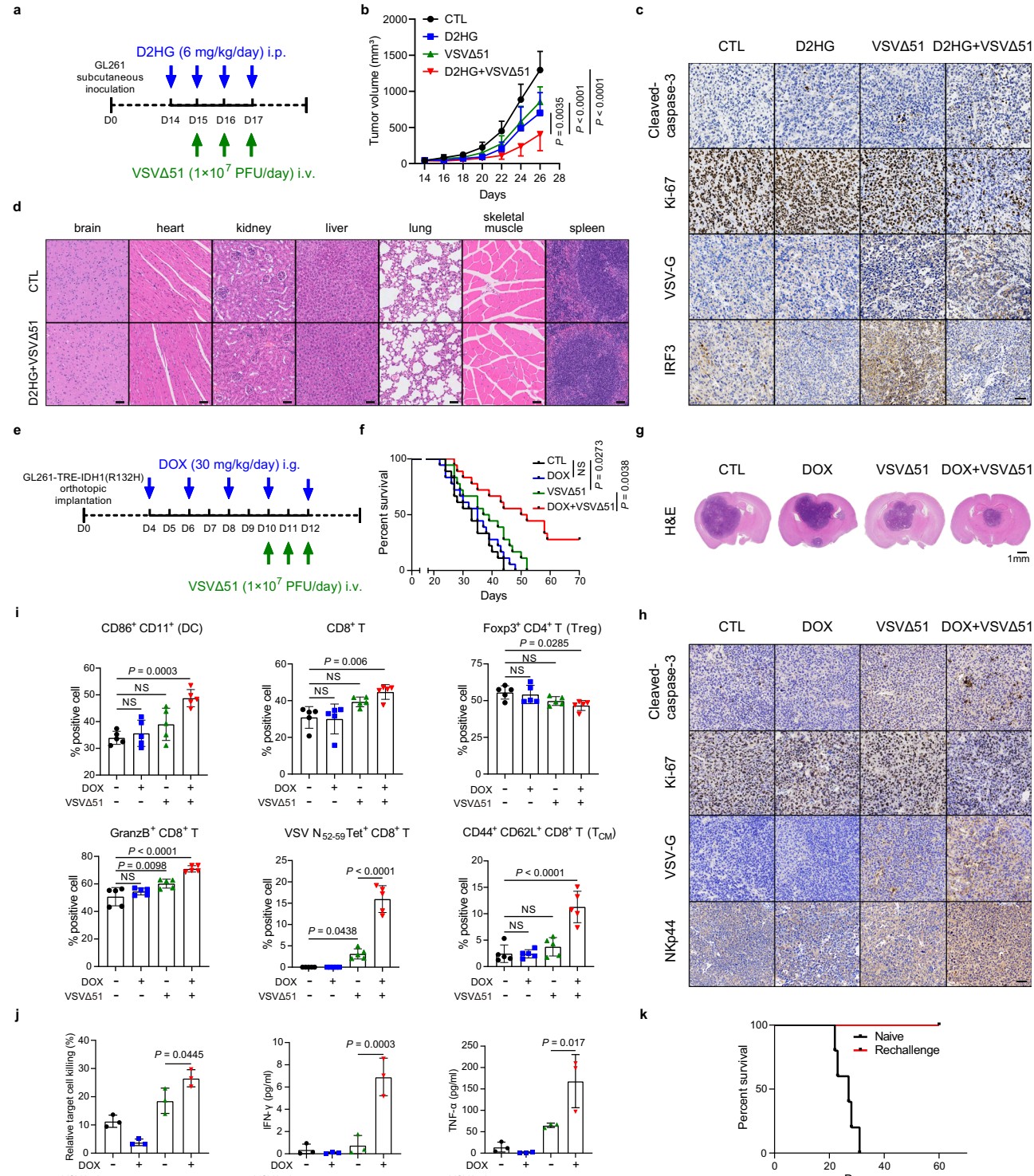

**Fig. 6 | Both D2HG and ectopic IDH1 mutant improve therapeutic efficacy of VSVΔ51 in vivo. a** Timeline of the experimental setup for **b**. **b** GL261 xenografts were treated with vehicle, VSVΔ51 (i.v., intravenous), D2HG (i.p., intraperitoneal) or a combination. $n = 5$ mice per group. **c** Tumor tissues from **b** were evaluated through immunohistochemistry staining for cleaved-caspase-3 (a marker of cell apoptosis), Ki-67 (a marker of proliferation), VSV-G, and IRF3. Representative images of $n = 3$. **d** H&E staining of vital tissues from **b**. Representative images of $n = 3$. **e** Timeline of the experimental setup for **f**. **f** Kaplan–Meier survival curve of the mice bearing intracranial GL261-TRE-IDH1(R132H) tumors were treated with vehicle, VSVΔ51 (i.v., intravenous), DOX (i.g., intragastric) or a combination. $n = 18$ mice per group. NS not significant. **g** H&E staining of mouse brain from **f** at day 24 after inoculation. **h** Immunohistochemistry analysis of cleaved-caspase-3, Ki-67, VSV-G and NKp44 (a marker of NK cells) in tumor tissues from **f**. Representative images of $n = 3$. **i** Percentages of DCs (CD86+ CD11+) among CD45+ cells, CD8+ T cells among CD3+ cells, Tregs (Foxp3+) among CD4+ T cells, and central memory T cells ($T_{CM}$, CD44+ CD62L+) among CD8+ T cells were analyzed in brain tumor tissues. The function of CD8+ T cells was evaluated by measuring granzyme B (GranzB). CD8+ T cells with TCR reactivity to the VSV $N_{52-59}$ H-2Kb immunodominant epitope were enumerated. $n = 5$ mice per group. NS not significant. **j** Splenic lymphocytes extracted from the mice in **f** were co-cultured with GL261 cells for 48 h. Cell viability of GL261 cells and cytokine secretion of the splenic lymphocytes were detected. $n = 3$ mice per group. **k** The survived mice ($n = 5$) from the combination experiment were rechallenged on day 60. Naive mice of similar age were implanted as controls ($n = 5$). Statistical significance was determined using two-way ANOVA in **b**, one-way ANOVA in **i**, **j**, or the log-rank (Mantel-Cox) test in **f**. Data represent the mean ± SD. Scale bar, 50 μm. Source data are provided in the Source Data file.

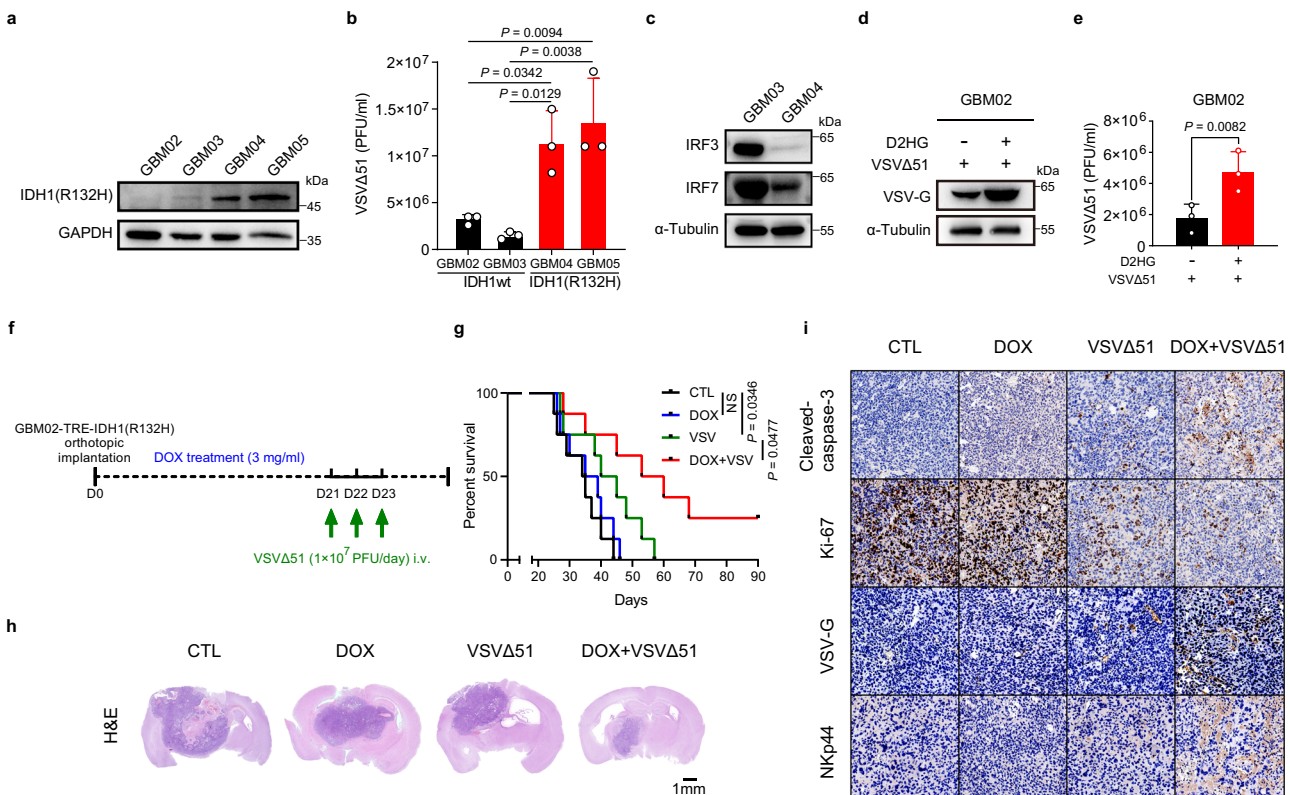

**Fig. 7 | IDH1 mutation renders patient-derived GBMs sensitive to VSVΔ51.**
**a** Western blot analysis of IDH1(R132H) in patient-derived GBM cells. **b** Patient-derived GBM cells were infected with VSVΔ51 (MOI = 1) for 12 h, and viral titers were determined. *n* = 3 biological replicates. **c** Western blot analysis of IRF3/7 in patient-derived GBM cells. **d**, **e** Cells were pretreated with or without D2HG (400 μM) for 48 h, followed by VSVΔ51–GFP (MOI = 0.01) infection for 24 h. **d** Viral protein VSV-G was analyzed by western blot. **e** Corresponding viral titers in supernatants were determined. *n* = 3 biological replicates. **f** Timeline of the experimental setup for **g**. **g** Kaplan–Meier survival curve of the mice bearing intracranial GBM02-TRE-

IDH1(R132H) tumors were treated with vehicle, VSVΔ51 (i.v., intravenous), DOX-containing drinking water or a combination. *n* = 8 mice per group. NS, not significant. **h** H&E staining of mouse brain from **g** at day 27 after inoculation. **i** Immunohistochemistry analysis of cleaved-caspase-3, Ki-67, VSV-G and NKp44 in tumor tissues from **g**. Representative images of *n* = 3. Statistical significance was determined using one-way ANOVA in **b**, two-tailed Student's *t*-test in **e**, or the log-rank (Mantel-Cox) test in **g**. Data represent the mean ± SD. Scale bar, 50 μm. Source data are provided in the Source Data file.

## RNA interference

Specific and scramble siRNAs were obtained from Ribobio (Guangzhou, China). Cell medium was replaced with 10% FBS in DMEM (without penicillin/streptomycin). SiRNAs were transfected using Lipofectamine RNAiMAX (13778-150, Thermo Fisher) with OPTI-MEM (31985070, Thermo Fisher). The siRNA sequences are described in Supplementary Table 1.

## Dual-luciferase reporter assay

HEK-293T cells were seeded in 24-well plates ($2 \times 10^5$ cells per well) and then transfected with 250 ng of the *IRF3* promoter dual-luciferase reporter plasmid together with empty vector (EV) or increasing amounts (wedge: 0.25, 0.5, and 1 μg) of plasmid encoding DNMT1 via Lipofectamine 3000 (Invitrogen). Twenty-four hours after transfection, the cells were stimulated with VSVΔ51 (MOI = 1) for another 12 h. The cells were lysed with passive lysis buffer and subjected to measurements of dual-luciferase activity with a Luciferase Reporter Assay System (Promega). Luciferase reporter activity was normalized to that of Renilla luciferase. The lysates left were collected for western blot analysis.

## Chromatin immunoprecipitation assay

ChIP assays were performed following the standard protocol of SimpleCHIP® Plus Sonication Chromatin IP Kit (56383, Cell Signaling Technology). Briefly, DMSO- and D2HG-treated LN-229 cells (~4 × 10⁶) were treated with 1% formaldehyde for 10 min at room temperature to

crosslink DNA and proteins. The reaction was terminated by the addition of the glycine solution and incubated at room temperature for 5 min. After cell lysis, the cross-linked chromatin was sonicated to an average size of ~500 bp and was immunoprecipitated with antibodies against DNMT1 and IgG. Purified ChIP DNA was amplified by qRT-PCR using specific primers targeting *IRF3* promoter, *IRF7* promoter, *TBK1* promoter and *IFNA1* promoter. Primers used for ChIP-qPCR are listed in Supplementary Table 1.

## Flow cytometry analysis

For multicolor flow cytometric analysis, brain tumor quadrants were harvested, minced, incubated with a Brain Tumor Dissociation Kit (130-095-942, Miltenyi), triturated, passed through a 70 mm screen, resuspended in FACS buffer (2% inactivated fetal calf serum in PBS), and stained with fluorochrome-conjugated anti-mouse antibodies from BioLegend. A Zombie Red Fixable viability kit (BioLegend) was used to stain dead cells. We followed a 'no-wash' sequential staining protocol (BioLegend) to stain dead cells and for surface staining. Intracellular Foxp3 staining was performed following the Foxp3 intracellular staining protocol (BioLegend). For single-color compensation controls, compensation particles (552845, BD) were used and stained with each fluorescently conjugated antibodies according to the manufacturer's instructions. For the Zombie Red assay, cells from the non-tumor and tumor quadrants, respectively, were used as single-color compensation controls. All samples were run in a Cytoflex flow cytometer (Beckman Coulter). Data were analyzed with CytExpert

software. Technicians acquiring and gating the data were blinded to the treatments.

## Isolation of splenic lymphocytes and cell co-cultures

GL261 cells were seeded in 96-well plates at 4000 cells per well in 0.1 ml medium. The next day, murine splenic lymphocytes were isolated using the Mouse Lymphocyte Separation Medium (7211011, Dakewe) according to the manufacturer's instructions. Then the appropriate number of splenic lymphocytes was added in each well on top of the adhered GL261 cells. The ratio of splenic lymphocytes (effector cells) to GL261 cells in co-cultures was 20:1. Forty-eight hours post co-cultures, the production and secretion of IFN-γ or TNF-α in cell supernatants was measured with a mouse IFN-γ ELISA kit (1210002, Dakewe) or a mouse TNF-α ELISA kit (1217202, Dakewe), and the viability of the adhered GL261 cells was detected by MTT.

## Statistical analysis

All statistical analyses were performed using GraphPad Prism software. Comparisons between different groups were made using Student's $t$-test or analysis of variance (ANOVA) as appropriate in the in vitro study. Values of tumor volume were analyzed by two-way ANOVA. The Kaplan–Meier curves were analyzed by log-rank test. All error bars indicate SD. Differences were considered significant if the $P$ value was <0.05.

## Reporting summary

Further information on research design is available in the Nature Portfolio Reporting Summary linked to this article.

## Data availability

The RNA-seq data used in this study have been deposited at the Gene Expression Omnibus under accession codes (GEO accession, GSE208634). The authors declare that the remaining data generated or analyzed during this study are available within the article, Supplementary Information, or Source Data file. Source data are provided with this paper.

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

## Acknowledgements

We thank Prof. Guangmei Yan (Zhongshan School of Medicine, Sun Yat-sen University), Prof. Qi Qi (School of Medicine, Jinan University), and Dr. Hao Duan (Sun Yat-sen University Cancer Center) for their technical assistance. This work was supported by grants from the National Natural Science Foundation of China (81972605 to H.Z. 82173829 to F.X.), the Guangdong Natural Science Funds of Distinguished Young Scholar (2021B1515020067 to H.Z.), and the Pearl River S&T Nova Program of Guangzhou (201906010069 to H.Z.).

## Author contributions

F.X. and H.Z. conceptualized the project. X.C., J.L., Yuqin L., F.W., X.L., F.X., and H.Z. developed methodologies. X.C., J.L., Yuqin L, Y.Z., F.W., Z.C., H.D., G.P., S.Y., Y.C., Q.L., X.S., Ying L., Z.Q., J.C., Y.H., X.W., Yuli L., X.L., and H.Z. performed the experiments and acquired the data. H.D., M.S., Y.Z., G.W., and K.L. contributed critical reagents and provided experimental support. X.C., J.L., X.L., F.X., and H.Z. conducted statistical analysis. X.C., J.L., and H.Z. wrote the manuscript, which was proofread and edited by all co-authors prior to submission. H.Z. supervised the work.

## Competing interests

The authors declare no competing interests.
