## [Peer Review File · Nature Communications]

Reviewers' Comments:

Reviewer #1:

Remarks to the Author:

This manuscript describes an interesting observation. IDH1 mutant GBM cells appear to be more susceptible to oncolytic virotherapy. This effect may be mediated by 2-HG, which is known to be substantially upregulated in IDH mutant tumors. 2-HG suppresses IRF3 that itself is regulated by DNMT1. They use a bunch of assays to substantiate this pathway, including CHIP and others. However, there are concerns with this study.

1. The paper relies entirely on established GBM cells. Essentially, all experiments should be repeated in relevant model systems of GBM. It is discouraged to express mutant IDH1 in established GBM lines. The biology of IDH wt and IDH mutant GBMs are very different.
2. The CHIP assay in figure 5C is not convincing and decisive. It should be repeated, e.g. with a read-out that involves real-time PCR analysis (with Ct values rather than this suboptimal semiquantitative approach the authors utilize).
3. Figure 6: They need to use a patient-derived IDH mutant GBM xenograft.
4. Figure 5L needs to include patient-derived IDH mutant and wild type lines.

Overall, the model systems used here are not ideal and may suggest a biology that is actually not relevant for IDH mutant glioma.

Reviewer #2:

Remarks to the Author:

IDH1 mutation impairs antiviral response and potentiates oncolytic virotherapy in glioma
Chen et al investigative biomarkers in the context of response to oncolytic virus therapy in glioma, with an emphasis on mutation status of IDH and DNMT1 expression. Overall the study is well designed to assess whether IDH1 mutation and its metabolite, 2-HG could affect the replication and oncolysis of oncolytic viruses. The authors demonstrate the mechanism behind mIDH1-stimulated virus replication in vitro. They confirmed their finding by utilizing publicly available patient datasets and human samples which make the study more relevant.

However, there are some concerns regarding the in vivo data. Addressing the revisions stated below will make this study more appealing.

Major revisions

1. The IRF3-dependent effect of 2HG on replication of VSV Δ 51 were documented in this study. Is this effect only observed with VSV Δ 51 virus or authors observed same effect with other oncolytic viruses? If they have not used other OVs, the title and abstract should be modified to indicate that this applies to VSV, since it is not known if other OVs (like DNA viruses) would also be affected
2. VSV Δ 51, like other oncolytic viruses, is shown to induce immunogenic cell death (ICD), and the induction of ICD activates immature DCs to transition to a mature phenotype with increased phagocytic function and ability to present antigens T cells, resulting in long-lasting protective anti-tumor immunity. Is there any synergistic effect observed on immunogenic cell death when combined with 2-HG? Do authors observe increased infiltration of DC population or activated microglia in TME? Did they re-challenge the surviving mice in combination group (Figure 6f) to assess whether long-lasting protective anti-tumor immunity is formed? What is the magnitude of CD8 T cells recall responses, effector, memory T cells ratio in these mice?
3. The authors stated that the effect of combination treatment is mediated by functional Granzyme B positive CD8 T cells in their orthotopic model. Are these CD8 T cells virus specific (viral antigen reactive) or tumor specific (mIDH tetramer positive CD8 T cells)? Did they check other immune

populations, such as NK cells in their immune competent model? Given to that they observed tumor growth in their flank model where they used immune deficient BALB/C nu/nu mice in which T cells are not functional, NK cell might influence controlling tumor cells as well.

Minor revisions

1. In line 50, 51 they state: "However, it has been very difficult to correlate viral activity with specific tumor-associated mutations"...but there have been papers that have done this...for example see PMID: 18345032
2. In figure 2 where the viral replication is measured, 2b and 2d for LN-229 do not reflect same pattern. 2b shows almost no GFP positive cells within 12h for both group and there is low number of GFP positive cells in control group for 24h whereas in 2d, virus titer is almost at same level with Dox treated group for 24h time points. Does author have any explanation for this? Is GFP signal not reflective of virus replication? Also, there is no statistical analyses provided for Figure 2d. Same for figure 2g, no GFP signals for only virus treated group, however titer analyses show indication of virus, although it is less than mIDH1 group. Showing whole tumor IF staining might be helpful to see the presence of virus.
3. In results section, it is indicated that time lapse was used for Figure 3b, however there is only one photo taken at a certain time point. For figure 3e, no p values are provided for other two groups, even though, the one-way ANOVA was used for analyses. Also, in same figure 12h were chosen to show the effect on virus replication, however as shown in figure 2a, replication of virus in both cell lines, LN-229 and LN-18, is limited within 12 hours. The effect might be related to infection efficiency rather than replication unless GFP signal is not a representation of virus replication.
4. In figure 6i, it is indicated that it shows CD8 ratio among CD45+ population, however, gating strategy shown in extended data indicates that it is the level CD8 population in CD3+ cells. This misleading info needs to be corrected.
5. There is no information provided in text part of result for cell lines, origin, mutation etc., they used or for usage of DOX which make it hard to follow the reasoning of experiment.
6. In extended Figure 1 legend, for ClueGO interaction analyses, no explanation is provided regarding the color and size of dots.

Reviewer #3:

Remarks to the Author:

The authors present a study examining the relationship between IDH1 mutations, D2HG, and the suppression of anti-viral immunity in pre-clinical models of glioma, including the response to oncolytic virotherapy in vivo. Overall, this is an important area of investigation, as oncolytic viruses are under clinical investigation for gliomas and the mechanisms of sensitivity or resistance are incompletely understood. The conclusion that IDH1 mutation, and associated 2-HG, potentiate VSV replication and oncolysis are well supported both in vivo and in vitro using multiple models. However, the proposed mechanism linking IRF3 expression to suppressed antiviral responses and enhanced oncolytic effects of VSV both in vitro and in vivo is not adequately supported.

Comments for the authors

How did the authors confirm the delivery of metabolites isocitrate, α KG, and D2HG in Fig 3a, 3b? Can the authors provide data on intracellular D2HG to correlate with viral replication effects? It is unclear if this was unmodified D2HG or octyl-D2HG, as is widely used.

It is important to specify the specific enantiomer of 2HG as D2HG throughout the manuscript.

The use of intraperitoneal injection of D2HG to test the hypothesis that tumor intrinsic IDH1-R132H and associated D2HG levels influence IRF3 signaling via DNMT1 is not well justified and subject to potential confounding factors of the delivered D2HG not impacting xenograft D2HG

levels or having unclear trans-acting effects on immune cells. To support the mechanistic conclusions drawn by the authors, the investigators should test that their delivered doses of D2HG lead to an increase in xenograft D2HG levels, or provide a discussion of the limitations associated with not having this measurement in the study.

The impact of D2HG on IRF3 and IFN α and IFN β induction suggest an anti-inflammatory effect, which is in line with multiple published studies on the role of D2HG. Can the authors discuss the relationship between this observation and the apparent potentiation of anti-tumor immunity observed in Fig 6I?

Additional details are needed on how IHC and IF data are quantified in ImageJ. Simply stating that these are analyzed in ImageJ is not sufficient. It is unclear if the data are analyzed on a cell-by-cell basis or simply the average pixel intensity over the whole image. Cell by cell would be appropriate.

Figure 1e-f: Based upon the proposed mechanism IRF3 expression should be downregulated after IDH1 induction, however, IRF3 is not shown despite other individual genes being shown.

Fig 4 While Figs 2 and 3 nicely demonstrate that D2HG and IDH1 mutations potentiate VSV replication using multiple cell lines and models, the conclusion that IRF3 control by IDH1 mutation is only demonstrated using one cell line. It is unlikely that all of the cell lines tested in Fig 3 actually mount an interferon response to VSV. At minimum the effect of D2HG on IRF3 and IFN responses as done in Fig 4 should be done in multiple models

Fig 4b-c: It is striking that IRF3 total levels are induced by VSV infection- generally IRF3 total levels are not induced during antiviral responses. Can the authors confirm that this is not p-IRF3? While blots are quantified, the apparent differences in IRF3, p-IRF3, MDA5 (an ISG), RIG-I (an ISG), and STAT1 (an ISG) are not convincing. If the bulk IFN response is impaired, it would be expected that overall ISG induction would also be impaired. However, this is not the case, particularly for RIG-I, MDA5, and STAT1. It may be more informative to probe for—and consider—IRF7, which controls IFN α induction (as opposed to beta) where there is a much more profound difference (Fig 4f vs g).

Fig 4d: total levels of IRF3 should be shown to indicate whether IRF3 overexpression occurs.

It is anticipated that the impact of D2HG from IDH1 mutations will broadly impact antiviral responses and other genes. To support the conclusion that enhanced VSV replication is IRF3 dependent, at minimum either blockade of TBK1-IRF3 signaling (BX795), antiviral responses (Ruxolitinib), or deletion of IRF3 should be performed to demonstrate that D2HG and/or IDH1 mutation potentiate VSV replication/oncolysis through modulating antiviral responses. Currently this work relies upon correlative and not definitive assessments. While Fig 5 does demonstrate enhanced binding of DNMT1 to the IRF3 promoter, this is anticipated to likely occur at many promoters across the genome due to increased D2HG and thus, once again, relies on correlation

Fig 6: It was not tested if GL261 cells mount different antiviral responses to VSV in vitro or in vivo. It is anticipated that IDH1 mutation and D2HG would mediate pleiotropic effects on both tumor cells and the TME. Thus, at minimum it should be tested if antiviral responses to GL261 occur in general, and if so, are impaired by D2HG/IDH1 mutation. Ideally the model would be tested in a IRF3 deleted GL261 +/- 2HG to determine if the effects of 2HG truly depend upon impaired IRF3 expression and thus, increased VSV oncolysis.

It is not clear how many times data were repeated/information about experimental repeats is not provided in the figure legends.

Minor comments

In introduction, can the authors clarify the sentence on lines 48-50 starting with "it is widely accepted that tumor selectivity...." it appears to be incomplete.

Rather than stating that cancer cells appear to lose their antiviral defense mechanisms, it would be helpful for the authors to more clearly delineate the types of immune and antiviral suppression mechanisms at play in gliomas, such as type I IFN loss at 9p21.

Point-by-point response to reviewers' comments

Manuscript: [NCOMMS-22-44407A-Z]

Reviewer #1

1. The paper relies entirely on established GBM cells. Essentially, all experiments should be repeated in relevant model systems of GBM. It is discouraged to express mutant IDH1 in established GBM lines. The biology of IDH wt and IDH mutant GBMs are very different.

Response: The reviewer mentioned that “the paper relies entirely on established GBM cells” and suggested us to repeat experiments in relevant model systems of GBM. This suggestion is crucial and fairly beneficial. Therefore, we carried out a series of experiments using patient-derived GBM cells to examine our findings. First, we observed that either ectopically expressed mutant IDH1 or D2HG can enhance viral replication in patient-derived GBM cells (Fig. 2f,g and Fig. 7d,e). Second, we found that IDH1mut GBM cells were more sensitive to VSVΔ51 infection than IDH1wt GBM cells (Fig. 7a,b). Furthermore, our data demonstrated that D2HG was able to enhance VSVΔ51 oncolysis in two patient-derived GBM cell lines (GBM01 and GBM02) (Extended Data Fig. 6). Mechanistically, doxycycline-induced IDH1(R132H) decreased the transcription of *IRF3/7* and downregulated the production of IFN- α and IFN- β in GBM02 cells (Fig. 4e,f). Finally, we orthotopically transplanted GBM02-TRE-IDH1(R132H) cells into mice (Fig. 7f). Consistent with our observation in GL261 xenograft model, doxycycline diet plus VSVΔ51 led to longer survival and effectively restricted tumor growth

compared with monotherapies alone (Fig. 7g-i). We have added these new data in Fig. 2f,g, Fig. 4e-f, Fig. 7, and Extended Data Fig. 6 in the revised manuscript.

Additionally, the reviewer stated that “it is discouraged to express mutant IDH1 in established GBM lines. The biology of IDH wt and IDH mutant GBMs are very different.” We quite agree with the reviewer’s opinion. Therefore, we analyzed VSV Δ 51 replication in IDH1wt and IDH1mut GBM cells. Our data showed that the capacity of viral replication in IDH1mut GBM cells was significantly stronger than that in IDH1wt GBM cells (Fig. 7a,b), suggesting that IDH1 mutant GBMs are more susceptible to OV infection. We have added these new data in Fig. 7a,b and modified the text to reference these new data on pages 12-13 lines 261-265 in the revised manuscript.

2. The CHIP assay in figure 5C is not convincing and decisive. It should be repeated, e.g. with a read-out that involves real-time PCR analysis (with Ct values rather than this suboptimal semiquantitative approach the authors utilize).

Response: We do agree with the reviewer’s opinion. Actually, the data shown in Fig. 5j were performed by ChIP-qPCR, and we think that these data were not clearly described in the previous version of our manuscript. Thus, we modified the description of these data in page 10 lines 198-201 to make it more clear for readers. Meanwhile, we have also provided the Ct values and calculation process in supplementary table 2.

3. Figure 6: They need to use a patient-derived IDH mutant GBM xenograft.

Response: We quite agree with this advice. Therefore, we investigated if mutant

IDH1 could enhance the susceptibility of patient-derived GBM xenograft to VSV Δ 51. We orthotopically transplanted GBM02-TRE-IDH1(R132H) cells into BALB/c-nu/nu mice, and animals were placed on a doxycycline-containing diet to induce IDH1(R132H) expression (Fig. 7f). Consistent with our observation in GL261 xenograft model in Fig. 6, doxycycline diet plus VSV Δ 51 led to longer survival and effectively restricted tumor growth compared with monotherapies alone (Fig. 7g,h). IHC staining results indicated that VSV Δ 51, in combination with doxycycline diet, (i) increased apoptosis, (ii) inhibited tumor cell proliferation, (iii) promoted viral replication, and (iv) enhanced infiltration of NK cells (Fig. 7i). We have added these new data in Fig. 7f-i and modified the text to reference these new data on page 13 lines 270-276 in the revised manuscript.

4. Figure 5L needs to include patient-derived IDH mutant and wild type lines.

Response: We appreciate this important suggestion. Therefore, we added three patient-derived GBM cell lines (GBM01, GBM02, and GMB03) into new Fig. 5l. Consistent with our previous results, the expression of DNMT1 correlates with enhanced cell killing by the combination treatment with VSV Δ 51 and D2HG in glioma cell lines and patient-derived GBM cell lines (new Fig. 5l,m). We have replaced Fig. 5l and 5m with the new data and modified the text to reference these new data on page 10 lines 208-212 in the revised manuscript.

Reviewer #2

Major revisions

1. The IRF3-dependent effect of 2HG on replication of VSV Δ 51 were documented in

this study. Is this effect only observed with VSV Δ 51 virus or authors observed same effect with other oncolytic viruses? If they have not used other OV, the title and abstract should be modified to indicate that this applies to VSV, since it is not known if other OVs (like DNA viruses) would also be affected.

Response: We appreciate this critical concern. In order to answer this important question, we chose two other well-recognized oncolytic viruses Zika virus (ZIKV) and Herpes Simplex Virus type I (HSV-1) to examine our findings. Our data showed that D2HG also significantly enhanced the replication of ZIKV and HSV-1 in glioma cells, thus indicating that our findings also apply to other oncolytic viruses (Extended Data Fig. 3). We have added these data in Extended Data Fig. 3 and modified the text to reference these new data on page 6 lines 128-131 in the revised manuscript.

2. VSV Δ 51, like other oncolytic viruses, is shown to induce immunogenic cell death (ICD), and the induction of ICD activates immature DCs to transition to a mature phenotype with increased phagocytic function and ability to present antigens T cells, resulting in long-lasting protective anti-tumor immunity. Is there any synergistic effect observed on immunogenic cell death when combined with 2-HG? Do authors observe increased infiltration of DC population or activated microglia in TME? Did they re-challenge the surviving mice in combination group (Figure 6f) to assess whether long-lasting protective anti-tumor immunity is formed? What is the magnitude of CD8 T cells recall responses, effector, memory T cells ratio in these mice?

Response: The reviewer pointed out that “VSVΔ51, like other oncolytic viruses, is shown to induce immunogenic cell death (ICD)” and questioned that “is there any synergistic effect observed on immunogenic cell death when combined with 2-HG?” In order to address this concern, we detected two hallmarks of ICD (CRT exposure and ATP secretion) in glioma cells treated with VSVΔ51, D2HG, or a combination. As our data showed, VSVΔ51 induced ICD in glioma cells, which could be further enhanced when combined with D2HG, suggesting that D2HG can promote ICD induced by VSVΔ51 in GBMs (Fig. 3k). We have added these new data in Fig. 3k and modified the text to reference these new data on page 7 lines 138-142 in the revised manuscript.

Furthermore, the reviewer mentioned that “the induction of ICD activates immature DCs to transition to a mature phenotype with increased phagocytic function and ability to present antigens T cells, resulting in long - lasting protective anti-tumor immunity” and asked that “do authors observe increased infiltration of DC population or activated microglia in TME? did they re-challenge the surviving mice in combination group (Figure 6f) to assess whether long - lasting protective anti-tumor immunity is formed? What is the magnitude of CD8 T cells recall responses, effector, memory T cells ratio in these mice?” To answer these critical questions, we repeated the animal experiments in Fig. 6f. Our new data showed that doxycycline diet plus VSVΔ51 treatment led to enhanced infiltration of DC cells (Fig. 6i). We further examined whether mice that received combination treatment could against a rechallenge with parental tumor cells. GL261-TRE-IDH1(R132H)

cells were implanted in the contralateral hemisphere of survivors from combination treatment group. Our data showed that mice that received combination treatment rejected rechallenged tumor (Fig. 6k). Finally, we observed that the combined treatment led to enhanced recall responses of central memory CD8⁺ T cells (T_{CM}) and had no impact on effector memory CD8⁺T cells (T_{EM}) (Fig. 6i and Additional Figure 1). Collectively, these results demonstrated that long-lasting protective anti-tumor immunity was formed after combination treatment. We have added these new data in Fig. 6i and 6k and modified the text to reference these new data on pages 11-12 lines 239-242 and page 12 lines 254-259 in the revised manuscript.

3. The authors stated that the effect of combination treatment is mediated by functional Granzyme B positive CD8 T cells in their orthotopic model. Are these CD8 T cells virus specific (viral antigen reactive) or tumor specific (mIDH tetramer positive CD8 T cells)? Did they check other immune populations, such as NK cells in their immune competent model? Given to that they observed tumor growth in their flank model where they used immune deficient BALB/C nu/nu mice in which T cells are not functional, NK cell might influence controlling tumor cells as well.

Response: The reviewer raised a crucial question concerning that “the authors stated that the effect of combination treatment is mediated by functional Granzyme B positive CD8 T cells in their orthotopic model. Are these CD8 T cells virus specific (viral antigen reactive) or tumor specific (mIDH tetramer positive CD8 T cells)?” We appreciate this critical concern. Actually, our data showed that doxycycline diet alone, which induced IDH1(R132H) expression, did not increase the percentage of

granzyme-B positive CD8⁺ T cells compared with control group (Fig. 6i). Moreover, a previous study has reported that the therapeutic efficacy of the IDH1(R132H) vaccine is dependent on CD4⁺ T cells rather than CD8⁺ T cells (Schumacher, et al., *Nature*, 2014). Based on this, we hypothesized that these CD8⁺ T cells might be virus specific. As expected, our data showed that doxycycline diet plus VSVΔ51 treatment led to increased percentage of VSV N₅₂₋₅₉ tetramer positive CD8⁺ T cells compared with VSVΔ51 treatment (Fig. 6i), indicating that the increased percentage of CD8⁺ T cells were virus specific. We have added these new data in Fig. 6i and modified the text to reference these new data on page 12 lines 244-247 in the revised manuscript.

In addition, the reviewer stated that “given to that they observed tumor growth in their flank model where they used immune deficient BALB/c nu/nu mice in which T cells are not functional, NK cell might influence controlling tumor cells as well” and suggested us to “check other immune populations, such as NK cells in our immune competent model.” We agree with the reviewer’s opinion and appreciate this suggestion. Therefore, NKp44 (a marker of NK cells) was examined in the orthotopic tumor sections by immunohistochemistry. As our data showed, dual treatment did lead to enhanced infiltration of NK cells in GL261 and patient-derived GBM xenografts compared with monotherapies (Fig. 6h and Fig. 7i). We have added these new data in Fig. 6h and Fig. 7i and modified the text to reference these new data on page 11 lines 235-236 and page 13 line 276 in the revised manuscript.

Minor revisions

1. In line 50, 51 they state: “However, it has been very difficult to correlate viral activity with specific tumor-associated mutations”...but there have been papers that have done this...for example see PMID: 18345032

Response: We agree with the reviewer’s opinion. We have rewritten this sentence and cited this important reference to express our view more appropriate on pages 3-4 lines 65-67 in the revised manuscript.

2. In figure 2 where the viral replication is measured, 2b and 2d for LN-229 do not reflect same pattern. 2b shows almost no GFP positive cells within 12h for both group and there is low number of GFP positive cells in control group for 24h whereas in 2d, virus titer is almost at same level with Dox treated group for 24h time points. Does author have any explanation for this? Is GFP signal not reflective of virus replication? Also, there is no statistical analyses provided for Figure 2d. Same for figure 2g, no GFP signals for only virus treated group, however titer analyses show indication of virus, although it is less than mIDH1 group. Showing whole tumor IF staining might be helpful to see the presence of virus.

Response: The reviewer mentioned that “in figure 2 where the viral replication is measured, 2b and 2d for LN-229 do not reflect same pattern. 2b shows almost no GFP positive cells within 12h for both group and there is low number of GFP positive cells in control group for 24h, whereas in 2d, virus titer is almost at same level with Dox-treated group for 24h time points.” We agree with the reviewer that these results are confusing. Thus, we repeated these critical experiments in Fig. 2b,

2c and 2e. Our new data suggested that the GFP signal data and the viral titer data reflected similar pattern. We have replaced Fig. 2b, 2c and 2e with the new data in the revised manuscript.

Moreover, the reviewer pointed out that “there is no statistical analyses provided for Figure 2e.” We appreciate the careful examination and have provided statistical analyses for new Fig. 2e. In addition, the reviewer stated that “same for figure 2g, no GFP signals for only virus treated group, however titer analyses show indication of virus, although it is less than mIDH1 group” and suggested that “showing whole tumor IF staining might be helpful to see the presence of virus.” Following the reviewer’s suggestion, we performed a global scan of the tumor IF staining section to show the presence of virus in vivo (Extended Data Fig. 2b). We have added these data in Extended Data Fig. 2b and replaced Fig. 2k with new representative images from the whole tumor IF staining.

3. In results section, it is indicated that time lapse was used for Figure 3b, however there is only one photo taken a certain time point. For figure 3e, no p values are provided for other two groups, even though, the one-way ANOVA was used for analyses. Also, in same figure 12h were chosen to show the effect on virus replication, however as shown in figure 2a, replication of virus in both cell lines, LN-229 and LN-18, is limited within 12 hours. The effect might be related to infection efficiency rather than replication unless GFP signal is not a representation of virus replication.

Response: The reviewer stated that “it is indicated that time lapse was used for

Figure 3b, however there is only one photo taken a certain time point.” We appreciate the careful examination. Actually, fluorescence microscopy was used for figure 3b, but not time-lapse microscopy. We have corrected this description on page 6 lines 125-127 in the revised manuscript. Moreover, the reviewer mentioned that “for figure 3e, no p values are provided for other two groups, even though, the one-way ANOVA was used for analyses.” We appreciate the careful examination and have provided statistical analyses for Fig. 3j.

In addition, the reviewer pointed out that “in same figure, 12h were chosen to show the effect on virus replication, however as shown in figure 2a, replication of virus in both cell lines, LN-229 and LN-18, is limited within 12 hours.” Actually, figure 3b showed the representative images of glioma cells infected with VSVΔ51 for 24 hours. We have added the description “24 hours post VSVΔ51-GFP infection” in Fig. 3b to make it more clear for readers.

4. In figure 6i, it is indicated that it shows CD8 ratio among CD45⁺ population, however, gating strategy shown in extended data indicates that it is the level CD8 population in CD3⁺ cells. This misleading info needs to be corrected.

Response: In figure 6i, it is actually CD8⁺ population among CD3⁺ cells as the reviewer mentioned. We appreciate the careful examination and have corrected the description in figure legend 6 in the revised manuscript.

5. There is no information provided in text part of result for cell lines, origin, mutation etc., they used or for usage of DOX which make it hard to follow the reasoning of experiment.

Response: Following the reviewer's suggestion, we have added the information of cell lines, origin, mutation to the methods section on page 15 lines 319-323.

6. In extended Figure 1 legend, for ClueGO interaction analyses, no explanation is provided regarding the color and size of dots.

Response: In terms of the reviewer's advice, we have provided an explanation regarding the color and size of dots for ClueGO interaction analyses in Extended Figure legend 1.

Reviewer #3

1. How did the authors confirm the delivery of metabolites isocitrate, α KG, and D2HG in Fig 3a, 3b? Can the authors provide data on intracellular D2HG to correlate with viral replication effects? It is unclear if this was unmodified D2HG or octyl-D2HG, as is widely used.

Response: We appreciate this critical concern. The design of isocitrate, α KG, and D2HG supplementation experiments in this study refers to the following literatures. Bullock et al showed that supplementation of isocitrate (IC) abrogated the erythroid iron restriction response in vitro (Bullock et al, *Blood*, 2010). In addition, Tseng et al demonstrated addition of α KG mediated a dynamic switch of glucose metabolism from glycolysis to oxidative phosphorylation in breast cancer cells (Tseng et al, *Cancer Research*, 2018). Yang et al revealed that exogenous D2HG inhibits TNF α -induced necroptosis in HT-29 cells (Yang et al, *Cell Reports*, 2018). Inspired by the reviewer's suggestion, we pretreated LN-229 cells with increasing doses of D2HG for 48 hours, and then detected the concentration of intracellular D2HG before

VSV Δ 51 treatment (Fig. 3h). Subsequently, D2HG-pretreated cells were infected with VSV Δ 51 for 24 hours, and the capacity of viral replication was determined (Fig. 3e-g). Our data demonstrated that intracellular D2HG levels correlated with viral replication (Fig. 3i). We have added these new data in Fig. 3e-i and modified the text to reference these new data on page 7 lines 132-136 in the revised manuscript.

In addition, the reviewer mentioned that “it is unclear if this was unmodified D2HG or octyl-D2HG, as is widely used.” We appreciated the careful examination. Actually, we used octyl-D2HG in this study. we have added a description of D2HG to the methods section on page 17 lines 381-382.

2. It is important to specify the specific enantiomer of 2HG as D2HG throughout the manuscript.

Response: We agree with the reviewer’s opinion and have specified the specific enantiomer of 2HG as D2HG throughout the manuscript.

3. The use of intraperitoneal injection of D2HG to test the hypothesis that tumor intrinsic IDH1-R132H and associated D2HG levels influence IRF3 signaling via DNMT1 is not well justified and subject to potential confounding factors of the delivered D2HG not impacting xenograft D2HG levels or having unclear trans-acting effects on immune cells. To support the mechanistic conclusions drawn by the authors, the investigators should test that their delivered doses of D2HG lead to an increase in xenograft D2HG levels, or provide a discussion of the limitations associated with not having this measurement in the study.

Response: We do agree with the reviewer's opinion. Following the reviewers' suggestion, we have provided a discussion of the limitations associated with not having this measurement on page 14 lines 301-304.

4. The impact of D2HG on IRF3 and IFN α and IFN β induction suggest an anti-inflammatory effect, which is in line with multiple published studies on the role of D2HG. Can the authors discuss the relationship between this observation and the apparent potentiation of anti-tumor immunity observed in Fig 6I?

Response: We appreciate this critical concern. As the reviewer mentioned, a previous study has reported an interesting finding that accumulation of D2HG leads to suppression of inflammatory pathways in the tumor microenvironment, thus resulting in a reduced recruitment of anti-tumor CD8⁺ T cells (Kohanbash et al., *J Clin Invest*, 2017). In this study, we observed that D2HG has ability to boost viral replication in glioma cells, resulting in direct lytic effect on tumor cells and induction of systemic antitumor immunity. Meanwhile, the enhanced replication of VSV Δ 51 can induce higher expression of proinflammatory molecules, such as calreticulin (Fig. 3k). We considered that D2HG might not be enough to reverse the antitumor immune responses caused by increased VSV Δ 51 replication, thereby the combination of D2HG and VSV Δ 51 exhibits a tipping of the immune balance in favor of anti-tumor immunity.

5. Additional details are needed on how IHC and IF data are quantified in ImageJ. Simply stating that these are analyzed in ImageJ is not sufficient. It is unclear if the data are analyzed on a cell-by-cell basis or simply the average pixel intensity over

the whole image. Cell by cell would be appropriate.

Response: We appreciate the careful examination. Actually, the IHC data were analyzed on a cell-by-cell basis as the reviewer mentioned. We have provided quantification details of IHC data in the methods section on page 19 lines 441-442.

6. Figure 1e-f: Based upon the proposed mechanism IRF3 expression should be downregulated after IDH1 induction, however, IRF3 is not shown despite other individual genes being shown.

Response: In order to address this concern, we re-analyzed the RNA-sequencing data shown in Figure 1c-e. However, we did not observe a significant downregulation of *IRF3* after doxycycline-induced IDH1(R132H) expression, whereas the target gene of IRF3, such as *BATF2*, was downregulated by mutant IDH1 (Additional Figure 2a). It is worth noting that *IRF3* expression was not increased in cells infected with VSV Δ 51 for 12 hours (Additional Figure 2a). Therefore, we detected the mRNA levels of *IRF3* in cells infected with VSV Δ 51 for different time points. Our data showed that the *IRF3* mRNA level reached peak at 6h post virus infection, which gradually returned to normal level from 12h to 24h (Additional Figure 2b). Considering that we chose cells infected with VSV Δ 51 for 12h to perform RNA-sequencing, which might make it difficult to discover the change of *IRF3* expression. Actually, we found that IDH1(R132H) can significantly decrease the transcription of *IRF3* at 6 hours post VSV Δ 51 infection (Additional Figure 2c).

7. Fig 4 While Figs 2 and 3 nicely demonstrate that D2HG and IDH1 mutations

potentiate VSV replication using multiple cell lines and models, the conclusion that IRF3 control by IDH1 mutation is only demonstrated using one cell line. It is unlikely that all of the cell lines tested in Fig 3 actually mount an interferon response to VSV. At minimum the effect of D2HG on IRF3 and IFN responses as done in Fig 4 should be done in multiple models.

Response: We do agree with the reviewer's opinion and have performed a series of experiments in multiple glioma cell lines to demonstrate the effect of IDH1 mutation on IRF3/7 and IFN antiviral responses. First, IDH1(R132H) downregulated the protein levels of IRF3 and IRF7 in GL261 cells (Extended Data Figure 4a). Moreover, IDH1(R132H) decreased the transcription of *IRF3/7*, downstream ISGs expression, and the production of IFN- α and IFN- β in GBM02, LN-18 and GL261 cells (Fig. 4e,f, Extended Data Fig. 4d,e). We have added these new data in Fig. 4e,f and Extended Data Fig. 4a, 4d,e and modified the text to reference these new data on page 8 lines 157-159 and lines 166-168 in the revised manuscript.

8. Fig 4b-c: It is striking that IRF3 total levels are induced by VSV infection- generally IRF3 total levels are not induced during antiviral responses. Can the authors confirm that this is not p-IRF3? While blots are quantified, the apparent differences in IRF3, p-IRF3, MDA5 (an ISG), RIG-I (an ISG), and STAT1 (an ISG) are not convincing. If the bulk IFN response is impaired, it would be expected that overall ISG induction would also be impaired. However, this is not the case, particularly for RIG-I, MDA5, and STAT1. It may be more informative to probe for—and consider—IRF7, which

controls IFN α induction (as opposed to beta) where there is a much more profound difference (Fig 4f vs g).

Response: We agree with the reviewer's opinion that the data of IRF3 total levels in Fig. 4b-c are confusing. Thus, we repeated these critical experiments using another IRF3 antibody with better specificity, and our new data showed that VSV Δ 51 infection has little effect on total IRF3 expression. Meanwhile, consistent with our earlier work, both IDH1(R132H) and D2HG downregulated the protein level of IRF3 in glioma cells (Fig. 4b,c).

Additionally, the reviewer concerned that "while blots are quantified, the apparent differences in IRF3, p-IRF3, MDA5 (an ISG), RIG-I (an ISG), and STAT1 (an ISG) are not convincing." We appreciate this critical concern and agree with the reviewer's opinion that "if the bulk IFN response is impaired, it would be expected that overall ISG induction would also be impaired. However, this is not the case, particularly for RIG-I, MDA5, and STAT1." Therefore, we repeated these experiments in Fig. 4b-c. As the reviewer stated, we observed that both IDH1(R132H) and D2HG downregulated the protein levels of RIG-I, MDA5, and STAT1 (new Fig. 4b,c). We have replaced Fig. 4b and 4c with the new data.

Furthermore, the reviewer suggested that "it may be more informative to probe for—and consider—IRF7, which controls IFN α induction (as opposed to beta) where there is a much more profound difference (Fig 4f vs g)." We appreciate this important suggestion. Therefore, we performed several critical experiments to investigate the effect of IDH1(R132H) on IRF7. First, we detected the protein level

of p-IRF7 and total IRF7 in glioma cells, and found that both IDH1(R132H) and D2HG suppressed total protein level of IRF7 (Fig. 4b,c). Second, exogenous IRF7 also abrogated the increased viral protein levels and titers induced by D2HG (Fig. 4d). Third, we found that IDH1(R132H) decreased the transcription of *IRF7* in cells infected by VSVΔ51 (Fig. 4e and Extended Data Fig. 4d). Furthermore, we observed that D2HG cannot further promote the replication in IRF7-knockdown glioma cells (Fig. 4h,j,l). Finally, we demonstrated that D2HG increased the association of DNMT1 with the *IRF7* promoter, thus inhibiting the transcription of *IRF7* (Fig. 4i,j). Collectively, these results suggested IRF7 to be another key regulator in the impaired antiviral responses induced by IDH1 mutation. We have added these new data in Fig. 4, Fig. 5 and Extended Data Fig. 4 in the revised manuscript.

9. Fig 4d: total levels of IRF3 should be shown to indicate whether IRF3 overexpression occurs.

Response: Following the reviewer's suggestion, we have provided total levels of IRF3 and IRF7 in new Fig. 4d.

10. It is anticipated that the impact of D2HG from IDH1 mutations will broadly impact antiviral responses and other genes. To support the conclusion that enhanced VSV replication is IRF3 dependent, at minimum either blockade of TBK1-IRF3 signaling (BX795), antiviral responses (Ruxolitinib), or deletion of IRF3 should be performed to demonstrate that D2HG and/or IDH1 mutation potentiate VSV replication/oncolysis through modulating antiviral responses. Currently this work

relies upon correlative and not definitive assessments. While Fig 5 does demonstrate enhanced binding of DNMT1 to the IRF3 promoter, this is anticipated to likely occur at many promoters across the genome due to increased D2HG and thus, once again, relies on correlation.

Response: We appreciate the reviewer's suggestion. Therefore, we knockdown IRF3/7 and found that D2HG cannot further promote the replication in IRF3-knockdown or IRF7-knockdown glioma cells (Fig. 4g-l). We have added these new data in Fig. 4g-l and modified the text to reference these new data on page 8 lines 168-170 in the revised manuscript.

11. Fig 6: It was not tested if GL261 cells mount different antiviral responses to VSV in vitro or in vivo. It is anticipated that IDH1 mutation and D2HG would mediate pleiotropic effects on both tumor cells and the TME. Thus, at minimum it should be tested if antiviral responses to GL261 occur in general, and if so, are impaired by D2HG/IDH1 mutation. Ideally the model would be tested in a IRF3 deleted GL261 +/- 2HG to determine if the effects of 2HG truly depend upon impaired IRF3 expression and thus, increased VSV oncolysis.

Response: The reviewer stated that "in Fig 6: It was not tested if GL261 cells mount different antiviral responses to VSV in vitro or in vivo. It is anticipated that IDH1 mutation and D2HG would mediate pleiotropic effects on both tumor cells and the TME" and suggested that "at minimum it should be tested if antiviral responses to GL261 occur in general, and if so, are impaired by D2HG/IDH1 mutation." We do agree with the reviewer's opinion. Therefore, we detected the expression of *IRF3/7*

and IFN-stimulated genes (ISGs) in GL261 cells. Our data showed that VSV Δ 51 infection significantly activated the antiviral responses in GL261 cells, while doxycycline-induced IDH1(R132H) can inhibit VSV Δ 51-induced activation of antiviral responses (Fig. 4e-f and Extended Data Fig. 4a, 4d). In addition, the reviewer suggested that “ideally the model would be tested in a IRF3 deleted GL261 +/- 2HG to determine if the effects of 2HG truly depend upon impaired IRF3 expression and thus, increased VSV oncolysis.” In terms of the reviewer's advice, we knockdown IRF3 in GL261 cells, and our data indicated that D2HG cannot further promote the replication in IRF3-knockdown cells (Fig. 4g,i,k). We have added these new data in Fig. 4 and Extended Data Fig. 4 in the revised manuscript.

12. It is not clear how many times data were repeated/information about experimental repeats is not provided in the figure legends.

Response: Following the reviewer's suggestion, we have stated the number of replicates in the figure legends.

Minor comments

1. In introduction, can the authors clarify the sentence on lines 48-50 starting with “it is widely accepted that tumor selectivity...” it appears to be incomplete.

Rather than stating that cancer cells appear to lose their antiviral defense mechanisms, it would be helpful for the authors to more clearly delineate the types of immune and antiviral suppression mechanisms at play in gliomas, such as type I IFN loss at 9p21.

Response: We agree with the reviewer’s opinion that “it would be helpful for the

authors to more clearly delineate the types of immune and antiviral suppression mechanisms at play in gliomas.” Following the reviewer’s suggestion, we have rewritten this sentence to introduce the relationship between the type I IFN loss at 9p21 and OV sensitivity in glioma on page 3 lines 62-63.

Additional Figure 1 for reviewer 2 concern 2

Additional Figure 1. Doxycycline diet plus VSVΔ51 treatment did not influence the effector memory T cells (T_{EM}) ratio in tumor tissues. Percentages of T_{EM} among CD8⁺ T cells were analyzed in brain tumor tissues. n = 5 per group.

Additional Figure 2 for reviewer 3 concern 6

Additional Figure 2. (a-c) Cells were pretreated with or without doxycycline (DOX) for 48 hours, followed by VSVΔ51–GFP infection (MOI = 1) for the indicated times. **(a)** RNA-seq analysis of LN229-TRE-IDH1(R132H) cells. **(b-c)** qRT-PCR assessing expression of *IRF3* mRNA. Data represent the mean ± SD. n = 3. ***P* < 0.01; ****P* < 0.001 by one-way ANOVA.

Reviewers' Comments:

Reviewer #1:

Remarks to the Author:

The authors present a revision of the manuscript that addresses some of the concerns raised previously.

However, there are still issues that need to be addressed:

Figure 7:

To be conclusive in vivo studies with GBM02, GBM03 vs GBM04 and GBM05 are necessary to support the claim that VSVdelta51 is more efficacious in the setting of an IDH1 mutation. While the in vivo model in 7f addresses some of the concerns it remains artificial and cannot substitute for the studies suggested above.

Figure 6A: The skin model is not meaningful in the context of GBM. The microenvironment of the skin is so much different from the one in the brain.

The CHIP assay is performed in LN229 cells which is a suboptimal model for GBM.

Figure 4b and 4c: Please show the PDX derived lines in lieu of LN229.

Figure 3 is concerning since it does for them most part contain established GBM lines which are not representative of the disease.

Overall, the revision is not completely satisfactory, and more works is still necessary for the paper to be a competitive candidate in a leading journal in the field.

Reviewer #2:

Remarks to the Author:

In the revised version of manuscript titled as IDH1 mutation impairs antiviral response and potentiates oncolytic virotherapy in glioma, Chen and colleagues made substantial changes to address our concern with the previous version.

Overall, the investigation into the potential impact of IDH1 mutation and its metabolite, 2-HG, on the replication and oncolysis of oncolytic viruses is well thought out and molecular mechanism behind increased oncolytic effect is deciphered. They used publicly accessible patient databases and human samples to corroborate their findings, which increased the study's applicability to patients. The link between IDH mutation and decreased anti-viral responses will be of interest to consider IDH status in gliomas management strategies. Moreover, the observed effect is not limited to VSV-G virus, which add a significant value for current clinical trials with different oncolytic viruses.

Reviewer #3:

Remarks to the Author:

The authors have addressed my concerns. I have no other concerns or issues with the manuscript as presented.

Point-by-point response to reviewers' comments

Manuscript: [NCOMMS-22-44407A-Z]

Reviewer #1

1. Figure 7: To be conclusive in vivo studies with GBM02, GBM03 vs GBM04 and GBM05 are necessary to support the claim that VSVdelta51 is more efficacious in the setting of an IDH1 mutation. While the in vivo model in 7f addresses some of the concerns it remains artificial and cannot substitute for the studies suggested above.

Response: We appreciate this critical concern and would like to answer this concern from two aspects. First, gliomas are often heterogenous, both phenotypically and with regards to gene expression (Stacey et al. *Trends in Neurosciences*, 2012). In addition to IDH1 mutation, gliomas harbor numerous other genetic mutations, such as TP53 and PTEN, which were also proven to affect antiviral responses (Larissa et al. *Trends in Cancer*, 2015). To avoid the interference of these genetic mutations, we decided to express mutant IDH1 in patient-derived IDH1wt GBM cells to determine its role in regulating antiviral response. Second, it is well recognized that gliomas with IDH1 mutations are difficult to culture and propagate in vitro (Piaskowski et al. *Br J Cancer*, 2011). Thus, glioma xenografts carrying IDH1 mutations are very scarce (Navis et al. *Acta Neuropathol Commu*, 2013). In spite of this, we agree with the reviewer that further study should be designed to compare the in vivo efficacies of oncolytic viruses in IDH1wt and IDH1mut xenografts. We have provided a discussion of this limitation on page 14 lines 301-303 in the revised

manuscript.

2. Figure 6A: The skin model is not meaningful in the context of GBM. The microenvironment of the skin is so much different from the one in the brain.

Response: The skin model shown in Figure 6A-D was designed to preliminary evaluate the efficacy of the combination treatment with VSV Δ 51 and D2HG, as well as the safety of this combination regimen. We do agree with the reviewer's opinion that "the microenvironment of the skin is so much different from the one in the brain". Therefore, we further determined our findings in two orthotopic glioma models (Fig. 6e-k, Fig. 7f-i).

3. The CHIP assay is performed in LN229 cells which is a suboptimal model for GBM.

Response: We agree with the reviewer's opinion. Therefore, we performed the ChIP assay experiments in GBM02 cells. Consistent with the results in LN-229 cells, D2HG increased the association of DNMT1 with *IRF3/7* promoters in GBM02 cells, whereas, it did not affect the affinity of binding between DNMT1 and the *TBK1* or *IFNA1* promoter (Supplementary Fig. 5f-g). We have added these new data in Supplementary Fig. 5f-g and modified the text to reference these new data on page 10 lines 197-201 in the revised manuscript.

4. Figure 4b and 4c: Please show the PDX derived lines in lieu of LN229.

Response: Following the reviewer's suggestion, we further explored the effect of D2HG on antiviral responses in patient-derived primary GBM cells. Our data showed that D2HG inhibited the upregulated expression of phosphorylated IRF3/7 induced by VSV Δ 51 in GBM02 cells. In line with our previous observations, we

found that D2HG downregulated the total protein levels of IRF3/7. We have added these new data in Supplementary Fig. 4b and modified the text to reference these new data on page 8 lines 159-160 in the revised manuscript.

5. Figure 3 is concerning since it does for them most part contain established GBM lines which are not representative of the disease.

Response: We appreciate this concern. In Figure 3, we chose 4 commonly used GBM cell lines (including LN-229, LN-18, LN2308 and GL261) to investigate if D2HG can enhance oncolytic virus replication and oncolysis. We also agree with the point that patient-derived primary GBM cells are more representative of the disease. Indeed, we demonstrated that D2HG was able to enhance the replication of VSV Δ 51 in GBM02 cells (Fig. 7d,e). Moreover, our data suggested that D2HG can promote ICD induced by VSV Δ 51 in GBM01 cells (Fig. 3k). Based on this, we think that the experiments performed in patient-derived GBM cells can further support our conclusion that D2HG enhances VSV Δ 51 replication and oncolysis in glioma.

Reviewer #2

1. In the revised version of manuscript titled as IDH1 mutation impairs antiviral response and potentiates oncolytic virotherapy in glioma, Chen and colleagues made substantial changes to address our concern with the previous version.

Overall, the investigation into the potential impact of IDH1 mutation and its metabolite, 2-HG, on the replication and oncolysis of oncolytic viruses is well thought out and molecular mechanism behind increased oncolytic effect is

deciphered. They used publicly accessible patient databases and human samples to corroborate their findings, which increased the study's applicability to patients. The link between IDH mutation and decreased anti-viral responses will be of interest to consider IDH status in gliomas management strategies. Moreover, the observed effect is not limited to VSV-G virus, which add a significant value for current clinical trials with different oncolytic viruses.

Response: We thank the reviewer for the careful and insightful review of our manuscript.

Reviewer #3

1. The authors have addressed my concerns. I have no other concerns or issues with the manuscript as presented.

Response: We thank the reviewer for the careful and insightful review of our manuscript.